# In vivo imaging and pharmacokinetics of percutaneously injected ultrasound and X-ray imageable thermosensitive hydrogel loaded with doxorubicin versus free drug in swine

Jose F. Delgado[1,2]*, Ayele H. Negussie[1], Nicole A. Varble[1,3], Andrew S. Mikhail[1], Antonio Arrichiello[1,4], Tabea Borde[1], Laetitia Saccenti[1], Ivane Bakhutashvili[1], Joshua W. Owen[1], Robert Morhard[1], John W. Karanian[1], William F. Pritchard[1‡], Bradford J. Wood[1,2‡]

**1** Center for Interventional Oncology, Clinical Center, National Institutes of Health, Bethesda, MD, United States of America, **2** Fischell Department of Bioengineering, A. James Clark School of Engineering, University of Maryland, College Park, Maryland, United States of America, **3** Philips Healthcare, Cambridge, Massachusetts, United States of America, **4** Department of Diagnostic and Interventional Radiology, UOS of Interventional Ra '1diology, Ospedale Maggiore di Lodi, Lodi, Italy

‡ These authors are joint senior authors on this work.
* jose.delgado@nih.gov

## Abstract

Intratumoral injections often lack visibility, leading to unpredictable outcomes such as incomplete tumor coverage, off-target drug delivery and systemic toxicities. This study investigated an ultrasound (US) and x-ray imageable thermosensitive hydrogel based on poloxamer 407 (POL) percutaneously delivered in a healthy swine model. The primary objective was to assess the 2D and 3D distribution of the hydrogel within tissue across three different needle devices and injection sites: liver, kidney, and intercostal muscle region. Secondly, pharmacokinetics of POL loaded with doxorubicin (POLDOX) were evaluated and compared to free doxorubicin injection (DOXSoln) with a Single End Hole Needle. Utilizing 2D and 3D morphometrics from US and x-ray imaging techniques such as Computed Tomography (CT) and Cone Beam CT (CBCT), we monitored the localization and leakage of POLDOX over time. Relative iodine concentrations measured with CBCT following incorporation of an iodinated contrast agent in POL indicated potential drug diffusion and advection transport. Furthermore, US imaging revealed temporal changes, suggesting variations in acoustic intensity, heterogeneity, and echotextures. Notably, 3D reconstruction of the distribution of POL and POLDOX from 2D ultrasound frames was achieved and morphometric data obtained. Pharmacokinetic analysis revealed lower systemic exposure of the drug in various organs with POLDOX formulation compared to DOXSoln formulation. This was demonstrated by a lower area under the curve (852.1 ± 409.1 ng/mL·h vs 2283.4 ± 377.2 ng/mL·h) in the plasma profile, suggesting a potential reduction in systemic toxicity. Overall, the use of POL formulation offers a promising strategy for precise and localized drug delivery, that may minimize adverse effects. Dual modality POL imaging enabled analysis of patterns of gel distribution and morphology, alongside of pharmacokinetics of local delivery. Incorporating hydrogels into drug delivery systems holds significant promise for improving

**Data Availability Statement:** All relevant data are within the manuscript and its Supporting Information files.

**Funding:** This work was supported by the Center for Interventional Oncology in the Intramural Research Program of the National Institutes of Health (NIH) by intramural NIH Grants NIH Z01 1ZID BC011242 and CL040015. J.F.D. was supported by the Jayne Koskinas Ted Giovanis Foundation for Health and Policy through the NIH Graduate Partnership Program from the Office of Intramural Training and Education (GPP-OITE). The funders had no role in study design, data collection and analysis, decision to publish, or preparation of the manuscript.

**Competing interests:** BW is Principal Investigator on the following CRADA's = Cooperative Research & Development Agreements, between NIH and industry: Philips (CRADA), Philips Research (CRADA), Celsion Corp (CRADA), BTG Biocompatibles / Boston Scientific) (CRADA), Siemens (CRADA), NVIDIA (CRADA), XAct Robotics (CRADA). ProMaxo (CRADA), Negotiating CRADA with Tempus, Galvanize, Theromics, Imactis, Canon Medical, Varian, MediView. NV is a staff Scientist from Philips Healthcare. This does not alter our adherence to PLOS ONE policies on sharing data and materials.

**Abbreviations:** POL, Poloxamer 407; DOX, Doxorubicin; SEHN, Single End Hole Needle; MSHN, Multiple Side Hole Needle; MPIN-ST, Multiple Pronged Injection Needle Short Tip; IM, Intramuscular; PEG, Polyethylene glycol; DSCPC, 1,2-diasteroyl-sn-glycero-3-phosphocholine; CT, Computed Tomography; US, Ultrasound; CBCT, Cone-beam CT; MBs, Microbubbles; UHPLC, Ultra-High-Performance Liquid Chromatography; INJ, Injection; AUC, Area Under the Curve; CV, Coefficient of Variation; Tmax, Time to reach maximum drug concentration; Cmax, Maximum drug concentration; $t_{1/2}\alpha$, Alpha half-life; $t_{1/2}\beta$, Beta half-life; CL/F, Apparent Clearance.

the predictability of the delivered drug and enhancing spatial conformability. These advancements can potentially enhance the safety and precision of anticancer therapy.

## Introduction

Image-guided local drug delivery enables the precise delivery of therapeutic agents to the tissue site of interest. This method differs from systemic delivery, which may harm healthy tissue and may not specifically target malignant tissue due to its broad distribution [1]. Intratumoral injection for administering anti-cancer drugs offers several benefits, including reduced systemic toxicities, lower drug doses, and increased local drug concentrations [2–7]. These advantages can potentially maximize the efficacy of treatment. Ultrasound (US), Computed Tomography (CT), or Cone-Beam CT (CBCT) can be used for image-guidance in intratumoral injections to improve on-target drug delivery [8–12]. However, intratumoral administration, even with image guidance, is still limited by variability in drug distribution post-injection and off-target leakage, unless a drug delivery vehicle is employed [12, 13].

The use of hydrogels for anti-cancer drug delivery offers several advantages, including providing sustained drug release, delivering high drug concentrations locally to enhance tumoricidal effects, and improving the predictability of drug deposition [13–30]. Optimizing injection parameters to enhance drug-loaded hydrogel delivery through percutaneous injections can potentially impact drug distribution and, consequently, therapeutic efficacy [9, 13, 14, 31, 32]. Poloxamer 407 (POL) is a thermosensitive hydrogel that has been widely used as drug delivery vector due to its capability to solubilize hydrophobic drugs [21, 22, 33–38]. POL is a tri-block co-polymer constituted by a hydrophobic block of polyoxypropylene in between two blocks of polyoxyethylene that form spatially ordered micelles at physiological temperature [21, 22, 33–38]. Highly ordered micelles allow the formation of in situ depots of POL in living tissues [22, 39–54]. For this reason, this tri-block co-polymer has been used to carry drugs for chemo- or immunotherapy, thereby improving therapeutic outcomes in animal models [22, 39–54].

However, there is a gap in knowledge pertaining to pharmacokinetics and drug distribution of drug-loaded thermosensitive hydrogels after percutaneous intratumoral delivery compared to free-drug injections. In a comparison of pharmacokinetic parameters between doxorubicin (DOX) solution and DOX loaded into a thermosensitive poly(organophosphazene) hydrogel in murine tumors, a significant dose accumulation of DOX was observed primarily in tumor tissues compared to free-drug injections [55]. In another study, pharmacokinetic parameters of a mixture of POL and Poloxamer 188 (a gelation temperature modulator) were evaluated for anti-parasitic drug delivery in a sheep model (49). Sustained drug release was observed in both *in vitro* and *in vivo* experiments, with the anthelmintic drug showing sustained release for up to 70 days in the *in vivo* setting. Additionally, the study demonstrated the capability to modulate the gelation temperature based on the polymer formulation. Moreover, pharmacokinetic analysis of the anthelmintic drug released from the poloxamer-based formulation indicated sustained release up to 70 days following administration. This suggests the potential for maintaining its antiparasitic efficacy post-administration of the poloxamer-drug formulation. Both studies demonstrated the capability of using thermosensitive hydrogels as *in situ* forming drug depots to avoid off-target toxicities and non-specific drug delivery as well as inducing *in vivo* sustained release. Exploiting the virtues of thermosensitive hydrogel properties as drug reservoirs make them ideal candidates for intratumoral injections.

Hydrogels for image-guided drug delivery can be prepared with a plethora of contrast agents to make them imageable. Some POL formulations have been used for transarterial embolization with the incorporation of iodine contrast agents [37] or for intratympanic delivery by the incorporation of microbubbles (MBs) as US contrast agents [21]. By harnessing the capabilities of POL formulations combined with US and x-ray imaging, monitoring drug delivery pre- to post-injection becomes feasible. This integration could be seamlessly incorporated into clinical workflows, akin to procedures such as tumor ablations or biopsies [56–60]. The ability to combine multimodal imaging with hydrogels for percutaneous injections in clinical workflows helps overcome the limitations pertaining to individual imaging techniques. Each imaging modality possesses unique strengths and intrinsic limitations[61–63]. US imaging strengths include that it is a less expensive modality than x-ray-based imaging, is not based on ionizing radiation which minimizes radiation exposure, and anatomical images can be observed in real time. One of the limitations of US visualization in needle-based procedures is the reliance on the surrounding tissue echogenicity, depth of target, size of the target, and interference of anatomical structures with the US beam [56]. The benefits of x-ray imaging, such as CT or CBCT, include the visualization and targeting of small lesions, easy visualization of needles, and in some scenarios, it can be considered qualitatively more spatially accurate [56, 58, 64, 65]. Limitations of CT imaging include the exposure to ionizing radiation, nephrotoxicity of iodinated contrast agents, metallic artifacts and the expense of equipment and maintenance compared to US [56, 58, 64, 65].

The use of multimodality imaging in intratumoral injections enables the implementation of navigation systems for image-guided interventions [62, 66–68] as the multiple imaging modalities can be co-registered and visualized in real-time, displaying needle positions and hydrogel depositions [61, 63, 69–71].

However, there exists lack of evidence regarding the *in vivo* delivery and specific distribution of imageable hydrogels using both US and x-ray guidance for various delivery devices. This study aims to address this knowledge gap by investigating the performance of needle devices in delivering an imageable hydrogel in a swine model. In addition, this study aims to leverage and characterize the imageability of POL with DOX and determine its short-term pharmacokinetic drug delivery parameters compared to free drug local injection.

## Materials and methods

All reagents and materials were obtained from commercial sources. The complete list of chemicals and reagents used during this study is included in supplemental information (**S1 Appendix**),

### Delivery needles

18G, 10cm Single End-Hole Needle (SEHN) and 19G, 7.5cm Multiple Side-Hole Needle (MSHN) were obtained from Cook Medical, Bloomington, IN, USA. 18G, 15cm, Multi-Pronged Injection Needle with Short Tip (MPIN-ST) was obtained from Rex Medical, Conshohoken, PA, USA.

### Microbubble (MBs) preparation

MBs were synthesized as previously reported [72] with slight modifications. 0.6mL of 25mg/mL of 1,2-distearoyl-sn-glycero-3-phosphocholine (18:0 DSPC) (Avanti lipids, Alabaster, Alabama, USA) and 0.4mL of 10mg/mL of Polyethylene glycol (PEG) 40 stearate, both in chloroform (Sigma Aldrich, Inc; Saint Louis, MO, USA), were mixed in an 8mL glass vial (Duran Wheaton Kimble, Millville, NJ, USA). The mixture was left to evaporate at 39°C in a hotplate

overnight to create a dry phospholipid film. The phospholipid film was solubilized with 2mL of 8:1:1 normal saline, glycerol, and polypropylene glycol solution (Sigma Aldrich) and heated at 79°C for 1h. The phospholipid solution was then ultrasonicated at 20kHz and 20% power for 2min with an ultrasonic probe (Q55 Qsonica Sonicators, Newtown, CT, USA) and then ultrasonicated for additional 20s at 20kHz and 90% power under perfluoro butane bubbling. Finally, the MBs were cooled in ice bath for immediate use.

## Preparation of ultrasound and x-ray imageable POL gel

POL gel was prepared as previously reported [37]. A 22% (w/v) formulation of POL (Sigma Aldrich) containing 40mg/mL of iodine from iodixanol (Visipaque 320 mg/mL, GE Healthcare, Waukesha, WI, USA) was solubilized in normal saline (110 g of POL, 62.5 mL of iodixanol, and 327.5 mL of normal saline) at 4°C under vigorous stirring for 24h. Similarly, POL containing iodixanol and 10mg/mL of DOX (LC Laboratories, Woburn, MA, USA) was prepared under the same conditions (27.5 mg of POL, 15.6 mL of iodixanol, 1.25 g of DOX, and 109.4 mL of normal saline) and stirred vigorously for 120h until complete dissolution was achieved. After complete mixture of POL with or without DOX, 0.01% or 0.1% MBs (v/v) (v/v is volume of MBs/volume of liquid POL) were added and gently manually shaken at 4°C until the formulation looked cloudy from the homogeneous dispersion of MBs. One-, and 10 microliters of MBs were used to obtain 0.01%, and 0.1% MBs (v/v), respectively in a total of 10 mL of POL solution containing or not containing DOX.

## Animal studies

Ten Yorkshire domestic female swine (Oak Hill Genetics, Ewing, IL, USA) weighting 52 – 61Kg were studied under a protocol approved by the National Institutes of Health Animal Care and Use Committee. Animals were sedated with intramuscular ketamine (25 mg/Kg), midazolam (0.5 mg/Kg), and glycopyrrolate (0.01 mg/Kg) and anesthetized with propofol (1 mg/Kg intravenous) and maintained under general anesthesia with isoflurane (1–5%, Isoflo, Abbott Animal Health, North Chicago, IL). The animals were intubated and mechanically ventilated during CT or CBCT acquisition. The right jugular vein was exposed, and a 6-F vascular sheath was placed. At the conclusion of the study, euthanasia was performed under general anesthesia by intravenous administration of Beuthanasia-D (pentobarbital sodium 390 mg/mL and phenytoin sodium 50 mg/mL).

## Optimization of %MBs in POL

A pre-procedural CT scan of the abdomen was conducted in anesthetized swine in supine position (Brilliance MX8000, Philips, Cleveland, OH); scan parameters: 120kVp and 328 mA, 2 mm slices at 1.5 mm intervals) to identify suitable injection sites in the liver with minimal vessel presence and safe distance from major arteries. Once the delivery sites were identified in the subject (n = 1), a SEHN was inserted percutaneously and advanced into the liver under US guidance (EPIQ 7, Philips, Cambridge, MA, USA) using a C5-1 curvilinear transducer. Doppler was used to identify any major arteries close to the needle tip. The other side of the SEHN was connected to a silicon tube (Smiths Medical, Dublin OH, USA) and attached to a syringe pump (Harvard Apparatus PhD Ultra Syringe Pump, Holliston, MA, USA) containing POL with 0.01% or 0.1% MBs. 4mL of POL containing 0.01% or 0.1% of MBs were injected at a rate of 10mL/h. Sequential CT images of POL were acquired at 0, 1,2,3, and 4mL post injection with breath holds. Two POL injections were performed with 0.01% MBs, and one with 0.1% MBs. After euthanasia, the liver was excised, and CT and US images (UI22, Philips) were additionally acquired.

3D volumetric distributions were segmented in 3D Slicer Software (3D Slicer, URL https://www.slicer.org/).

## Needle performance study under CT and US guidance

**Determination of 3D distribution of POL across three needle devices.**   POL containing iodine (40 mg/mL) and MBs (0.1%, v/v) was delivered percutaneously to the liver in three swine under the following parameters: Swine 1: 52Kg, SEHN (4 mL; 10 mL/h; 4 different sites); Swine 2: 53 Kg, MPIN-ST (4 mL; 100 mL/h; 3 different sites), and Swine 3: 58 Kg, MSHN (2 mL; 10 mL/h; 3 different sites). Additional injections with MPIN-ST (5 mL; 100 mL/h; 3 different sites) were performed intramuscularly in the paraspinous muscles in one of the subjects. Sequential CT scans were obtained in all animals to acquire 3D POL distribution in tissue. In 10 mL/h injections, sequential CT images of deposited material in tissue were acquired at 0, 6, 12, 18, and 24 minutes. Using MPIN, the prongs were initially deployed to cover a diameter of 2 cm and during injection were gradually retracted to a final diameter of 0.5 cm in 0.5 cm intervals without rotation of prongs. The volume, surface area, and Hounsfield units of POL 3D distributions were quantified using 3D Slicer Software by intensity thresholding (different density levels). Sphericity and solidity of 3D distributions were calculated using the following equations as previously reported [73].

$$Sphericity = \frac{\pi^{\frac{1}{3}}(6V)^{\frac{2}{3}}}{SA} \qquad (Eq1)$$

$$Solidity = \frac{Volume}{Convex\ hull} \qquad (Eq2)$$

Where V and SA represent volume and surface area, respectively. The convex hull—defined as the smallest polygon containing the whole segmented volume [74]—was calculated by exporting the 3D distributions as STL files from 3D Slicer to Blender Software (Blender, URL https://www.blender.org/).

Additionally, a manual injection of 2 mL of POL was performed in one kidney and analyzed for 3D morphometrics (volume, surface area, sphericity, solidity) in one animal weighing 52 kg.

**Multimodal analysis of post-injection POL distribution: Integrating CT, ultrasound, and 3D mapping techniques.**   After POL injection, three Hounsfield Units thresholds were defined for segmentation, corresponding to three relative iodine concentration ranges, 7–13, 13–27, and 24–40 mg/mL. To estimate the relative iodine concentration within POL deposited after injection, a linear relationship between Hounsfield units and iodine concentration was assumed as described in reference [9, 14], with a maximum iodine concentration of 40 mg/mL. Color-coded contour plots were generated using MATLAB (R2020a version, Mathworks, Natick, MA) to illustrate areas of relative iodine concentration.

To determine the 1D distribution of relative iodine concentration in tissue, a line was drawn along the coronal plane and pixel intensities in CT images of POL injected with three needle devices were recorded using Fiji Software [75] (Fiji, URL: https://imagej.net/software/fiji/downloads). The Area Under the Curve (AUC) for the plots of relative iodine concentrations versus normalized distance (where the longest distance of the selected region of interest was defined as 1 unit) along the coronal plane of the POL deposition in tissue was calculated using GraphPad Prism 9,2 (GraphPad, Boston, MA, www.graphpad.com). Additionally, MATLAB was used to compute area, major axis (equatorial), minor axis (meridian), circularity, and solidity using CT slice selected bisecting the 3D gel distribution in the coronal plane.

Fiji Software was employed to determine lengths or areas of injections in cm or cm$^2$, respectively.

B-mode images from US underwent similar processing as described for CT imaging to extract morphometric data including perimeter.

Three-dimensional US imaging was conducted using an electromagnetic-based tracking system (EPIQ 7 PercuNav, Philips) with a C5-1 transducer. The US transducer location was tracked and swept over POL deposits. US image planes were converted to DICOM and oriented in 3D space using MATLAB. Segmentation and morphometric analysis of the resulting US volumes were performed using 3D Slicer, following procedures analogous to those used for CT imaging analysis. This included measurements such as sphericity, volume, and solidity in 3D.

For SEHN needles, serial CT imaging was conducted at various progressive injection volumes (pre-injection and at 1, 2, 3, and 4 mL) and at the same needle position. Each mL of POL was segmented accordingly. STL files were exported for each segmented object, and POL depositions per mL were spatially aligned using the needle as a reference in Meshmixer. The 3D coordinates were extracted from the aligned volumes using MATLAB, and these coordinates were used to plot the 3D distribution of POL per mL.

## Short-term *in vivo* imaging and pharmacokinetic study of POL-based formulation with DOX, and formulation without DOX

**Administration of POLDOX and DOXSoln formulations.** Two cohorts of swine were used in this experiment. The first cohort (n = 3, body weight 52, 54, and 55 Kg), received a percutaneous injection of 4 mL of POL containing iodine (40 mg/mL), MBs (0.1% v/v), and 10 mg/mL of DOX (POLDOX) using a SEHN needle. The second cohort (n = 3, body weight 52, 57, and 55 Kg) received a percutaneous injection of 4 mL of DOX at 10 mg/mL with iodine (DOXSoln) using a SEHN needle. Both percutaneous injections were performed at a rate of 10 mL/h. Needle placement and injection were guided using US and CBCT imaging to ensure accurate targeting and placement of the injection within liver tissue.

**Temporal and morphometric analysis of POLDOX formulation distribution in tissue.** CBCT (Allura Xper FD20, Philips, Best, the Netherlands; scan parameters: 120kVp and 148 mA) was employed to capture images of POLDOX and DOXSoln formulations at different timepoints during and after their administration. From CBCT images, 3D morphometric parameters, including sphericity and solidity were extracted using 3D Slicer. The spatial and temporal distribution of the injected materials per mL were examined by tracking their deposition and concentration within the tissue. Segmentation of the needle and the deposited materials within tissue using 3D Slicer software provided detailed visualization of their spatial distribution and interaction with the surrounding tissue. Fluoroscopic images were also obtained to visualize the distribution of POLDOX and DOXSoln. For temporal deposition analysis, CBCT images were acquired at 0, 6, 12, 18, and 24 mL during POLDOX administration, and 0, 30, 60, 90, 120, 180, and 240 min after administration. The volume percentage of POLDOX from the segmented volume was calculated with 3D Slicer post-injection assuming a 100% volume at 0 min post-injection. This calculation allowed quantification of the percentage of volume occupied by the injected material within the region of interest (ROI) over time. Additionally, sphericity and solidity measurements were obtained from the segmented volumes of POLDOX, providing further insights into its shape, structure, and interaction with the surrounding tissue and vessels. MATLAB Software was used to extract 2D morphometrics from CBCT such as area, circularity, and solidity measurements at these timepoints.

US B-mode imaging was conducted at 0, 30, 60, 90, 120, 180, and 240 min after POLDOX administration. Entropy, defined as a statistical measure of randomness used to characterize

texture of US images, was calculated with MATLAB. Acoustic heterogeneity was assessed by calculating pixel standard deviations within manually defined regions of interest at POLDOX deposition, using Fiji software. Uniform frequency and gain settings were maintained for all image acquisitions. Similarly, CBCT scans were employed to assess relative iodine concentrations in POLDOX at the same time intervals. The scans were analyzed to depict areas of three ranges of relative iodine concentrations, creating color-coded contour plots using MATLAB. MATLAB codes and Image J methodology can be found in the supplemental information (**S1 Appendix**), The area under the curve (AUC) for each relative iodine concentration over time was determined using GraphPad Prism 9.2.

**Plasma collection and storage.**   Blood (2 mL with 35 μL of heparin) from the central venous line was collected at 0.5, 2, 5-, 10-, 20-, and 24-min during administration, and subsequently at 2-, 5-, 10-, 20-, 40-, 60-, 90-, 120-, 180-, and 240-min after administration. Blood samples were centrifuged at $1.5 \times 10^3$ rpm for 10 minutes at 4°C using a Legend Micro17R centrifuge (Thermo Scientific, Waltham, MA, USA) to obtain plasma, which was stored at -80°C for further analysis.

Plasma obtained from a euthanized swine without treatment was used to prepare calibration standards.

**Tissue collection and storage.**   After the last time point for blood collection, swines were euthanized as described above, and liver, heart, kidney, and spleen were excised. The administration site in the liver was isolated from the rest of the organ using image-guidance, sliced into 5 mm pieces, and placed in histological cassettes (Leica Biosystems, Deer Park, IL, USA). These slices were then flash-frozen in 2-mercaptoethanol (Sigma Aldrich) pre-cooled in liquid nitrogen. Additionally, sections of the liver (far from the administration site), heart, kidney, and spleen were sliced and stored in 50 mL conical tubes at -80°C for DOX quantification. Tissues obtained from a euthanized swine without treatment were excised and used to prepare calibration standards for DOX quantification.

**Chromatographic equipment and conditions.**   HPLC analysis of plasma and tissue samples were performed as previously reported [76] with slight modification.

The LC system used was an Agilent 1290 Infinity Ultra-High-Performance Liquid Chromatography (UHPLC) system equipped with an Infinity binary pump, Infinity autosampler, Infinity Thermostated Column Compartment (TCC), and a 1260 Infinity Fluorescent Detector (FLD). Data acquisition and processing were conducted using LC ChemStation software.

Column separation employed an Agilent Zorbax Rapid Resolution HT Eclipse Plus C18 column (1.8 μm, 4.6 mm × 50 mm), using a binary mobile phase system of water (0.1% trifluoroacetic acid)/acetonitrile (0.1% TFA) with a volume ratio of 64:34 for plasma and 72:28 for tissue analysis. The elution flow rate was set at 1.5 mL/min for a runtime of 1.5 minutes for plasma samples and 1.0 mL/min for 10 minutes for tissue samples. Detection utilized the 1260 Infinity FLD with excitation/emission wavelengths of 480/593 nm.

DOX was detected at 0.58 minutes in plasma and 3.5 minutes in tissue samples, while Daunorubicin (DNR) an internal standard (IS), was detected at 0.97 minutes in plasma and 5.2 minutes in tissue samples.

## Plasma and tissue DOX quantification

**Preparation of calibration standards for plasma.**   Stock solutions of DOX (0.2–100 μg/mL) and IS DNR (25 μg/mL) were prepared in deionized water and stored at 4°C in polypropylene conical tubes. For sample preparation, 90 μL of plasma containing 10 μL of DNR (25 μg/mL) and 10 μL of DOX solution at various concentrations (0–50 μg/mL) were incubated at 37°C for 15 minutes. Then, 20 μL of aqueous $KH_2PO_4$ solution (20 mM, pH 3.8) was

added, and the mixture was further incubated at 37˚C for 15 minutes to extract analytes. Subsequently, 10 µL of saturated zinc sulfate and 10 µL of acetone were added and incubated at 37˚C for 15 minutes to precipitate proteins. The samples were then centrifuged at 17 x g at 4˚C for 1.5 minutes using a Legend Micro17R centrifuge (Thermo Scientific). After centrifugation, 100 µL of the supernatant was transferred to another vial, and the solvent was removed using a TurboVap LV (Caliper Life Sciences, Inc, Hopkinton, MA, USA) at 40˚C under a stream of compressed air. The dried residue was completely dissolved in 100 µL of Ultra High Performance Liquid Chromatography (UHPLC) mobile phase, and 90 µL was transferred into a UHPLC vial. Of this solution, 15 µL was injected into the UHPLC system for analysis, as described previously.

**Plasma sample preparation for treatment groups.** Plasma samples (100 µL), each containing DNR (10 µL of 25 µg/mL) as internal standard, were incubated for 15 minutes at 37˚C. The samples were then processed as described above.

**Tissue samples preparation.** Tissue samples were homogenized in aqueous $KH_2PO_4$ solution (20 mM, pH 3.8) at a concentration of 25 mg/mL using a Bead Mill Homogenizer (Omni, Kennesaw, GA, USA) in a 30 mL prefilled bed tube (Omni, USA). The homogenized tissue samples were kept at -80˚C until further analysis.

**Preparation of calibration standards for tissue.** Stock solutions of DOX (0.2–100 µg/ml) and the IS DNR (25 µg/mL) were prepared in de-ionized water and stored at 4˚C in polypropylene conical tubes. Calibration standards were prepared by spiking blank tissue homogenate (270 µL, 25 mg /mL, n = 3) with 30 µL of DOX (0.2–100 µg DOX/mL) and 30 µL of DNR (25 µg/mL) solutions, followed by incubation at 37˚C for 15 min to facilitate protein-drug binding. The samples were then mixed with 250 µL acetone and 120 µL $ZnSO_4$ solutions (saturated) and re-incubated at 37˚C for another 15 min to precipitate the proteins. Subsequently, the samples were centrifugally filtered with Eppendorf Centrifuge 5417 R, 14000 rpm at 4˚C for 15 min. Four hundred microliters of supernatant were transferred to another vial and evaporated using TurboVap® LV (Caliper life sciences) at 40˚C under a stream of compressed air. Prior to UHPLC analysis, the dried residue was completely dissolved in 150 µL of UHPLC mobile phase, centrifuged, and 100 µL was transferred into a UHPLC vial. From this solution, 10 µL was injected into the UHPLC system for analysis, as described previously.

**Tissue sample preparation for treatment groups.** IS solution (30 µL, 25 µg DNR /mL) was added to 300 µL of the sample (n = 3, 25 mg /mL) in a 1 mL vial. The vials were mixed for 5 seconds and incubated at 37˚C for 15 minutes. The samples were then processed using the same procedure as the calibration standards of blank tissue samples (see above).

**Data analysis.**

1. - DOX concentration. The retention times of peaks from sample analytes were identified based on the chromatographic profile of a mixture of pure DOX and DNR. Concentrations of the analyte in tissue and plasma samples were quantified using peak-area ratios of the sample analyte to the IS from the linear calibration curve, established using least squares regression method based on nominal concentration. Drug concentration in various tissues were calculated as the ratio of DOX concentration to mg of tissue used. For the determination of DOX concentration from organ samples, a single-point calibration method is used.

2. - Calculation of pharmacokinetic parameters. Pharmacokinetic parameters including maximum plasma drug concentration ($C_{max}$) and the time ($T_{max}$) needed to reach $C_{max}$, were determined from DOX plasma concentration-time curves. Due to our experimental design, we administered each formulation (4 mL) over a period of 24 minutes. Blood sampling was performed during the administration period (from t = 0 to t = 24 minutes) and during the post-administration period (from t = 24 to t = 264 minutes).The area under the curve

(AUC) was calculated for the administration time ($AUC_{0-24min}$) and post-administration time ($AUC_{24-264min}$) using the trapezoid rule (Eq 3). The total AUC ($AUC_{total}$) is defined as the sum of $AUC_{0-24min}$ and $AUC_{24-264min}$ (Eq 4) [77]:

$$AUC = \frac{(C_{n-1} + C_n)(t_n - t_{n-1})}{2}. \tag{Eq3}$$

Where C is concentration of drug in plasma, and t is time.

$$AUC_{total} = AUC_{administration} + AUC_{post-administration} \tag{Eq4}$$

Apparent clearance (CL/F) for POLDOX and DOXSoln formulations was estimated with the following equation [77]:

$$CL/F = \frac{Dose}{AUC_{tot}}, \tag{Eq5}$$

where F is the fraction of the dose that reaches the general circulation. In extravascular administration, not all of the drug enters systemic circulation; therefore, clearance is expressed as a value that is dependent on bioavailability. Rate constants for the distribution and elimination phases were determined by plotting the logarithm of DOX concentrations (post-administration) in plasma against time (from 4mL post-administration time, 24 min, to 264 min) and the slope is determined from linear regression and its multiplied by 2.303 to take into account the base 'e' to obtain the distribution and elimination rate constants. Subsequently, half-life ($t_{1/2}$) of DOX distribution ($t_{1/2\alpha}$) and elimination ($t_{1/2\beta}$) half-lives were calculated using the equations below [78]:

$$t_{1/2} = \frac{Ln2}{\alpha}, \tag{Eq6}$$

where $\alpha$ is the constant rate at the distribution phase.

$$t_{1/2} = \frac{Ln2}{\beta}, \tag{Eq7}$$

where $\beta$ is the constant rate at the elimination phase.

## Statistical analysis

A one-way ANOVA with post-hoc Tukey test was conducted to assess statistical differences between groups. T-tests were performed for comparisons within two groups. GraphPad Prism 9.2 was utilized for data visualization, plots, and statistical analyses.

## Results

### Optimization of %MBs in POL

The visibility of percutaneous POL injections with MBs under US and x-ray imaging was evaluated *in vivo* (**S1A-S1H Fig in S1 Appendix**). Injections of 4 mL POL with 0.01% MBs exhibited a hypoechoic core surrounded by a hyperechoic halo under US (**S1C Fig in S1 Appendix**) with acoustic intensity of 16.6 (a.u.). However, they were less readily visible compared to injections with 0.1% MBs. In contrast, injections of 4 mL POL with 0.1% MBs showed an increased acoustic intensity of 75.3 (a.u.) and enhanced POL visibility under US. Qualitatively, the morphology of POL observed from US and CT images appeared elliptical. The long axis measurements from ultrasound (US) (1.1 cm and 1.8 cm) and 2D CT coronal plane (1.4 cm and 2.3

cm) of explanted swine liver (**S1E, S1G, and S1H Fig in** S1 Appendix) were similar to the measurements obtained in vivo, 3 hours post-injection (1.4 cm x 1.8 cm) (**S1C, and S1F Fig in** S1 Appendix). Furthermore, leakage of POL noted in the explanted liver corresponded to observations made in vivo (**S1C and S1F Fig in** S1 Appendix).

### Needle performance study under CT and US guidance

**3D CT imaging morphometrical analysis.** 3D distribution of POL injected with three needle devices is depicted in Fig 1. Injection of POL with SEHN (Fig 1A–1C) and MPIN injected intramuscularly (MPIN (IM)) (Fig 1J–1L) showed minimal vessel leakage, whereas MSHN (Fig 1D–1F) and MPIN (Fig 1G–1I) exhibited a high degree of leakage. The reproducibility of injected volumes with different needle devices is presented in Fig 2. with specific values of 3D morphometric values detailed in **S1 Table in** S1 Appendix.

**Multimodal analysis of post-injection POL distribution: Integrating CT, ultrasound, and 3D mapping techniques.** Volumetric measurements from CT imaging indicated that SEHN and MPIN injections were close to 4 mL and 5 mL of MPIN (IM) respectively (Fig 2A). Comparing the 3D POL distribution with SEHN and MPIN injections revealed that MPIN injection had a higher surface area in the liver (p = 0.0082) (Fig 2B). SEHN injections

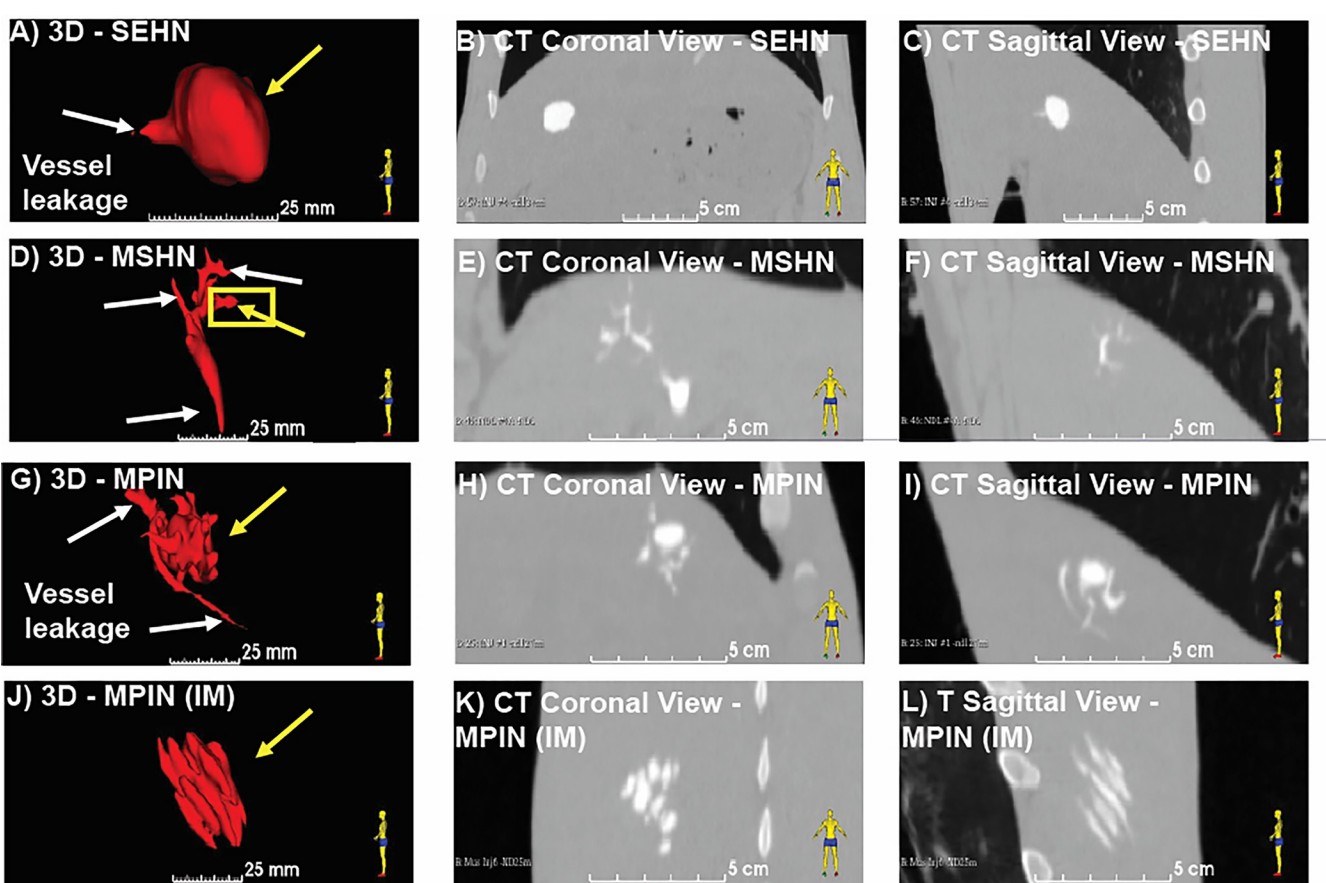

**Fig 1. 3D distribution of POL injected in swine with three needle devices.** (A) Volumetric distribution of POL injected in the liver with SEHN (n = 4) including CT images of coronal (B) and sagittal planes (C). (D) Volumetric distribution of POL injected in the liver with MSHN (n = 3) including CT images of coronal (E) and sagittal planes (F). (G) Volumetric distribution of POL injected in the liver with MPIN (n = 3) including CT images of coronal (H) and sagittal planes (I). (J) Volumetric distribution of POL injected with MPIN intramuscularly [MPIN (IM)] (n = 3) including CT images of coronal (K) and sagittal planes (L). White arrows indicate vessel leakage, and yellow arrows indicate localized injection.

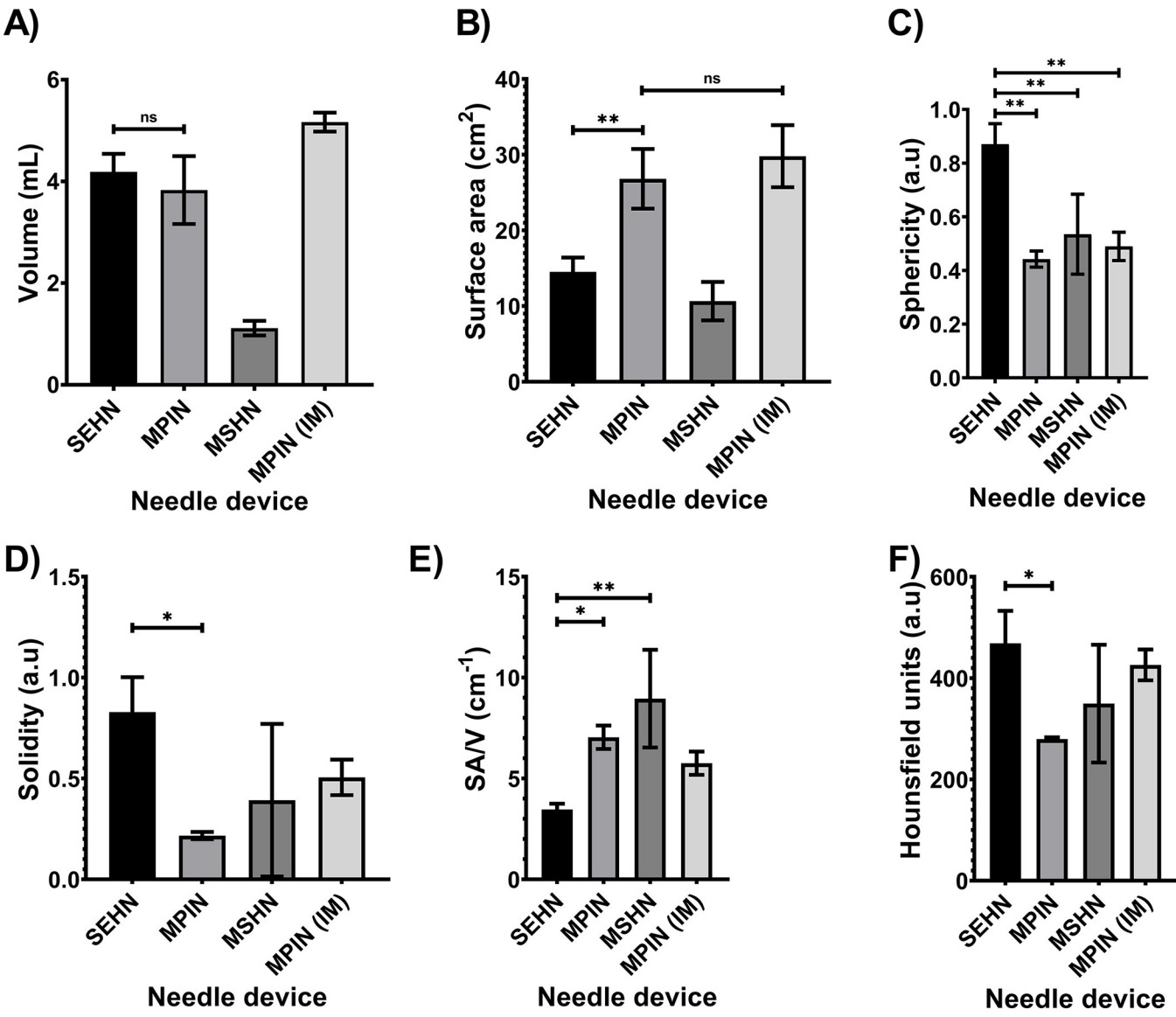

**Fig 2. 3D morphometrical and radiopacity characterization of POL injections and needle devices.** (A) Volume, (B) surface area, (C) sphericity, (D) solidity, (E) SA/V, and (F) radiopacity. Error bars depict standard deviations. Filled bars correspond to mean measurements: SEHN (n = 4), MSHN (n = 3), MPIN (n = 3), and MPIN (IM) (n = 3). Not-significant (ns, p>0.05), *p<0.05, and **p<0.01, from one-way ANOVA statistical test. SA/V, surface area to volume ratio.

exhibited higher sphericity (p = 0.0008) (**Fig 2C**) and solidity compared to other needle devices (p = 0.0037) (**Fig 2D**). MPIN showed a higher surface-area-to-volume (SA/V) ratio compared to SEHN (p = 0.0007) (**Fig 2E**). The average radiodensity within the distribution were similar across all injections except for SEHN and MPIN (p = 0.0378) (**Fig 2F**). Surface area, sphericity, and SA/V for MPIN and MPIN (IM) were similar (p = 0.4137, p = 0.2481, and p = 0.0534 respectively) (**Fig 2B–2D**), but solidity differed (p = 0.0051) (**Fig 2D**).

3D distributions of POL injected into renal tissue (S2 **Fig in S1 Appendix**) were also analyzed for morphometrics, resulting in two localized injections and one injection with leakage. The measurements for these injections in renal tissue were as follows: Volume = 1.3 ±0.7 mL; surface area = 7.9 ±2.3 $cm^2$; sphericity = 0.8 ±0.4; and Solidity = 0.6 ±0.4.

**Table 1. 2D morphometrics of POL depositions injected with MSHN, and MPIN from CT imaging coronal planes.**

| Needle device | Area (cm$^2$) | | | Circularity (a.u) | | | Solidity (a.u) | | |
|---|---|---|---|---|---|---|---|---|---|
| | INJ.1 | INJ. 2 | INJ. 3 | INJ. 1 | INJ. 2 | INJ. 3 | INJ.1 | INJ.2 | INJ.3 |
| MPIN | 0.4 ±0.4 | 0.6 ±0.5 | 0.2 ±0.3 | 0.7 ±0.4 | 0.6 ±0.3 | 0.9 ±0.1 | 0.8 ±0.3 | 0.8 ±0.1 | 0.9 ±0.0 |
| MSHN | 0.6 ±0.2 | 0.5 | 0.3 ±0.1 | 0.5 ±0.2 | 0.8 | 0.5 ±0.2 | 0.7 ±0.1 | 0.9 | 0.8 ±0.1 |
| MPIN (IM) | 2.3 | 1.5 ±1.9 | 0.9 ±1.1 | 0.2 | 0.7 ±0.4 | 0.6 ±0.3 | 0.5 | 0.9 ±0.1 | 0.9 ±0.1 |

± depicts standard deviations of mean values. Mean values varied depending on number of depositions detected from each injection (INJ).

**2D CT imaging morphometrical analysis.** The analysis of SEHN POL injections in the coronal 2D plane resulted in a single gel deposition area measuring 2.6±0.5 cm$^2$, with major (equatorial) and minor (meridian) axes measuring 2.3 ± 0.2 cm and 1.5 ± 0.1 cm, respectively (**S3A Fig in S1 Appendix**). SEHN injections exhibited a more consistent gel distribution compared to the other needle devices, which resulted in multiple areas of deposition (**S3B-S3D Fig in S1 Appendix**). MPIN (IM) injection typically showed either a single large deposition or two areas, with one notably larger than the other (**S3D Fig in S1 Appendix**). The circularity and solidity of SEHN injection were 0.9±0.01 a.u and 1.0±0.00 a.u, respectively (**S3E Fig in S1 Appendix**). MPIN, and MSHN needle devices resulted in a variety of irregular shapes evidenced by the lower circularities and solidities presented in most of the cases (**S3F-S3G Fig in S1 Appendix**). Within the areas of deposition in MPIN (IM), several small and interconnected beads (circularity = 0.7±0.3 a.u; solidity = 0.9±0.1) were detected with an average area in each single bead of 0.1±0.0 cm$^2$, and major and minor axes of 0.6±0.1 cm, and 0.3±0.0 cm, respectively (**S3H Fig in S1 Appendix**). **Table 1**. summarizes 2D morphometrics obtained after analysis of CT images of coronal planes for all needle devices.

**2D CT imaging iodine distribution analysis.** SEHN demonstrated consistent results regarding the major area of concentrated iodine (in the 27 to 40 mg/mL range) within the injection site (**Fig 3A**) compared to other needle devices (**Fig 3B–3D**).

MSHN showed areas containing a broad range of relative concentrations, with a high degree of non-localized and leaked material compared to other needle devices (**Fig 3B**). POL injections with MPIN needle resulted in homogenous areas of relative iodine concentration which corresponded to the needle tips as sources of infusion (**Fig 3C**). The area with maximum range of relative iodine concentrations (27 to 40 mg/mL) post-injection with SEHN was bigger than lower ranges of concentrations (7 to 13 mg/mL; p = 0.0016, and 13 to 27 mg/mL of iodine; p = 0.0017) (**Fig 3E**). Overall, areas with relative iodine concentration ranges were not significantly different within comparisons of each needle device (p>0.2) (**Fig 3E**). **S2 Table in S1 Appendix** summarizes the areas of relative iodine concentration values depicted in **Fig 3E**.

2D analyses were performed on POL injections to assess predictability across three needle devices in the coronal plane using CT imaging. AUC of the relative concentration profiles over normalized distance was calculated for each needle device (**S4A-S4D Fig in S1 Appendix**). Normalization of distance was used to make the AUC under the curve for each injection of POL comparable. SEHN and MPIN (IM) were identified as the injection devices with less variability (in terms of AUC) compared to other needle devices (**S4E Fig in S1 Appendix**). The AUC of iodine concentration profiles for SEHN and MPIN (IM) had lower standard deviation compared to MSHN and MPIN (**Table 2**). SEHN had lower standard deviation compared to MSHN and MPIN. MPIN (IM) and SEHN exhibited the lowest CV of AUC, indicating greater consistency and reproducibility. MPIN and MSHN had the highest CV, suggesting higher variability in predictability (**Table 2**). The AUC of SEHN and MPIN injections were found to be statistically different (p = 0.0125). These results indicate that SEHN and MPIN (IM)

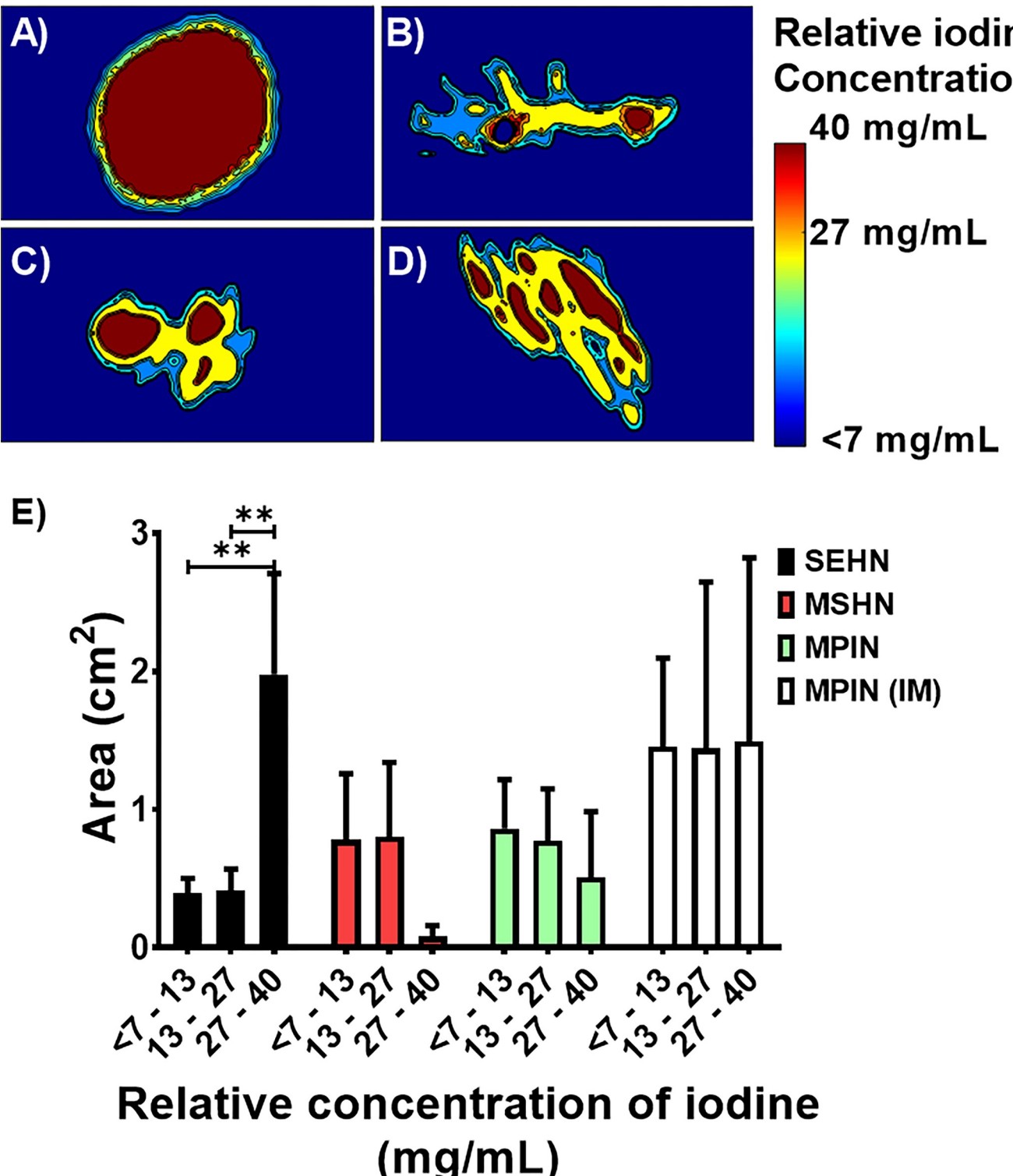

**Fig 3. Color-map of cross-sectional area of relative iodine concentration ranges for three needle devices after injection of POL.** (A) SEHN, (B) MSHN, (C) MPIN, and (D) MPIN (IM). Error bars represent standard deviations, while filled bars represent the mean of the areas. (E) Distribution of areas of relative iodine concentration ranges for three needle devices after injection of POL. SEHN (n = 4), MSHN (n = 3), MPIN (n = 3), and MPIN (IM) (n = 3). **p<0.01, from one-way ANOVA statistical test.

**Table 2. AUC values of relative iodine concentration over normalized distance in tissue.**

| Needle device | AUC (mg/mL*normalized cm) | n | CV |
|---|---|---|---|
| SEHN | 37.0 ±0.7 | 4 | 1.9 |
| MSHN | 34.0 ±3.1 | 3 | 9.0 |
| MPIN | 31.6 ±1.7 | 3 | 5.5 |
| MPIN (IM) | 34.5 ±0.1 | 3 | 0.4 |

± Depicts standard deviations. CV = Coefficient of variance.

demonstrate higher predictability in terms of AUC compared to other needle devices. Additionally, they show lower variability in AUC measurements, particularly MPIN (IM) and SEHN.

**US imaging and 2D comparison to CT imaging.** After the pilot study, a formulation of POL with 0.1% microbubbles (MBs) was used because this concentration provided hyperechoic depositions of POL, as shown in **S1D Fig in S1 Appendix**. **Fig 4** incorporated distributions of POL from US imaging pre- and post-injection. SEHN POL injections exhibited clear US imageability with specific dimensions: an area of 2.3±0.3 cm, a major axis of 2.4±0.3 cm, and a minor axis of 1.4±0.2 cm (**Fig 4A**). MSHN POL injections were characterized by an elliptical shape that remained visible under US immediately post-injection (**Fig 4B**). Visibility of POL injected with MPIN in the liver was challenging, with some depositions appearing hypoechoic and hyperechoic, making it difficult to detect the entire treated area under US (**Fig 4C**). Two out of three injections could be analyzed using US imaging.

POL injected into muscular tissue with MPIN (IM) resulted in an irregular deposition visible under US, although US contrast was not readily visible. Acoustic artifacts from swine ribs also affected US visibility (**Fig 4D**).

**S5 Fig in S1 Appendix** shows hypoechoic depositions of POL with SEHN and MPIN (IM) after liver and paraspinous muscle tissue explantation. However, visualizing injections under US after tissue explantation was challenging for MSHN and MPIN due to iodine clearance.

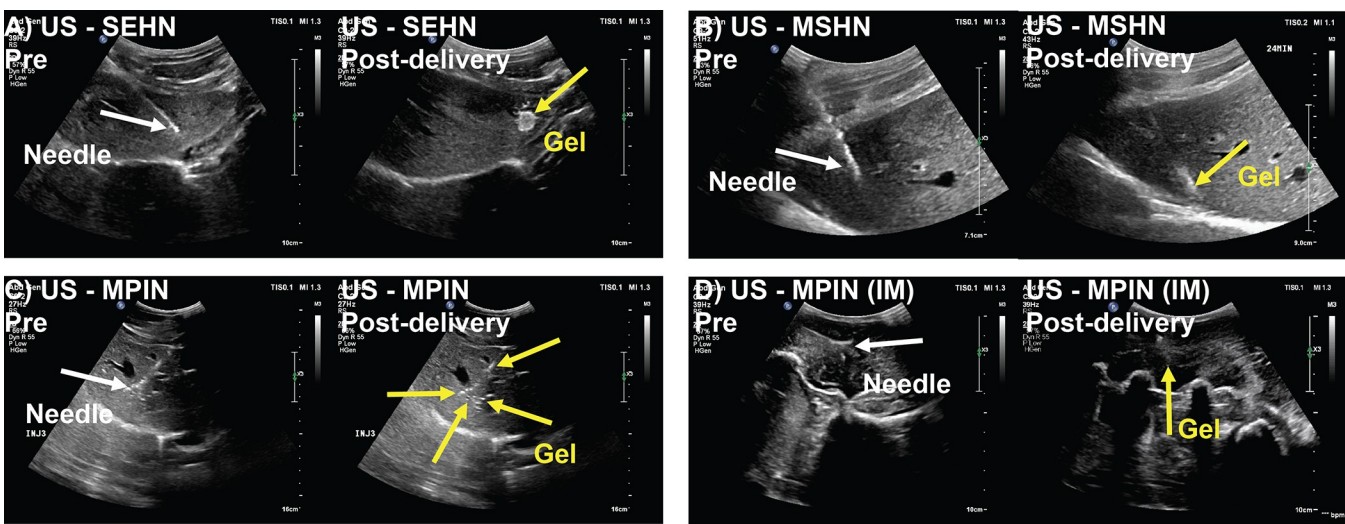

**Fig 4. US B-mode images of POL injected for three needle devices pre- and post-delivery.** n = 3 for each needle device. White arrows indicate the needle and yellow arrows indicate the gel.

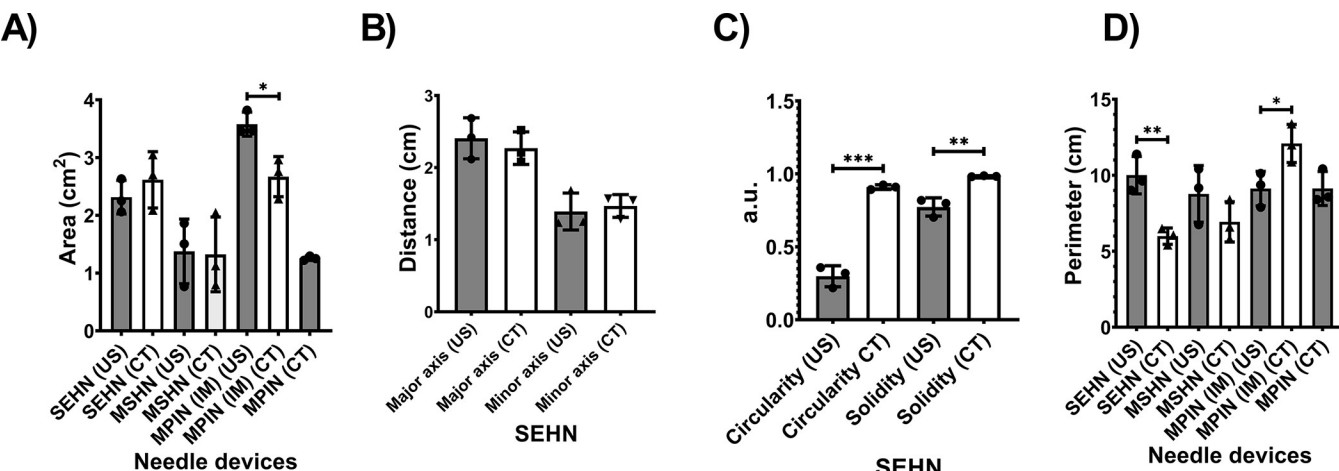

**Fig 5. Comparison of 2D morphometrics of POL depositions from US and CT imaging for three needle devices (n = 3).** Error bars represent standard deviations of mean value. *p<0.05, **p<0.01, ***p<0.001, and ****p<0.0001, from t-test statistical test.

The analysis of 2D distributions of POL post-injection was obtained from ultrasound (US) and CT imaging across different needle devices (**Fig 5**). Overall, comparison of US and CT imaging for areas from each needle device were not significantly different, except for POL deposited with MPIN (IM) (p = 0.0178) (**Fig 5A**). Measurements of major and minor axes of POL deposition with SEHN were not different (**Fig 5B**). However, US and CT data revealed that circularities and solidities of deposited POL with SEHN were significantly different (p = 0.0001 and p = 0.0043, respectively) (**Fig 5C**). In addition, perimeters of deposited POL using SEHN and MPIN (IM) were different (p = 0.0066 and p = 0.0385, respectively) when compared each needle with its respective US and CT imaging (**Fig 5A**).

These findings indicate that there are variations in the areas, circularities, solidities, and perimeters, across different needle devices when POL deposited using the three needles (**S3 Table in S1 Appendix**).

**US imaging and 3D comparison to CT imaging in SEHN.** Due to the simplicity in morphology of POL injected with SEHN (one spherical deposition), a 3D analysis from US and CT imaging modalities was performed for this needle device (**Fig 6**). The comparison of the volume obtained from US and CT were not statistically different (p = 0.0624) (**Fig 6A**), nevertheless, the average of volume computed from US was higher with higher standard deviations compared to CT. The sphericities of POL obtained from US and CT were different (p = 0.0469) (**Fig 6B**), however, the comparison of solidities resulted in no differences (p = 0.1592) (**Fig 6C**). The surface-area-to-volume ratio (SA/V) presented differences (p = 0.0083) (**Fig 6D**).

The 3D distribution of POL from US imaging (n = 3) was obtained and depicted in **Fig 6E**. From These results, US 3D POL distribution deposition was localized with an elliptical shape without substantial leakage (**Fig 6E**). During the registration of 2D frames from US imaging, the procedure was easily performed with 0.1% MBs. Additionally, he US imageability was studied for deposited POL in tissue without DOX and resulted in a decrease of acoustic intensity from 148.7 ±23.5 a.u to 112.4 ±17.0 a.u over approximately 1.5h with no differences (p = 0.0967) (**Fig 6F**). **S4 Table in S1 Appendix**. summarizes the 3D analysis from US and CT imaging modalities.

**Serial imaging of POL injection with SEHN.** The 3D distribution of POL injected with the SEHN needle device indicated localized distribution of injected POL.

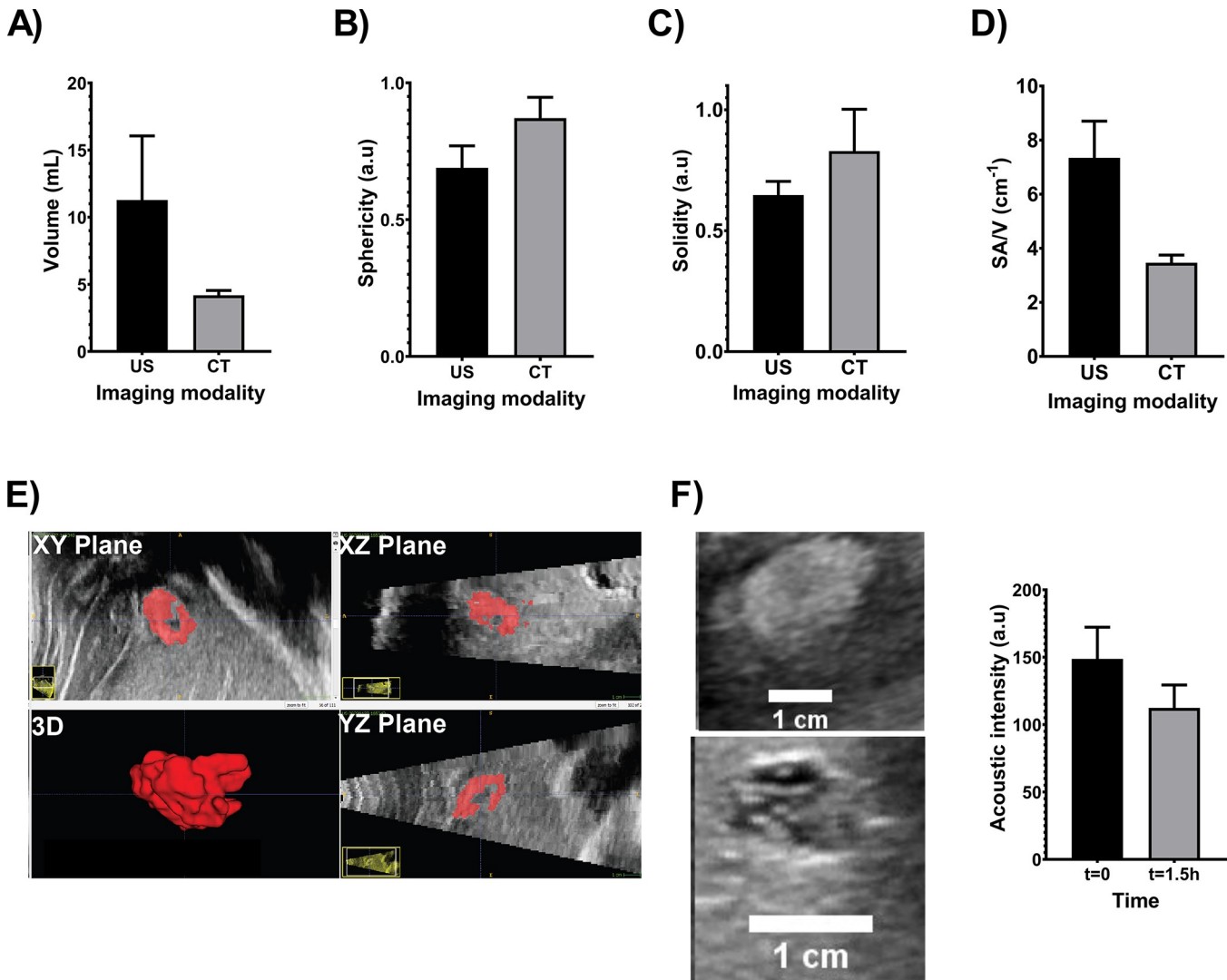

**Fig 6. 3D morphometrical comparison of POL injected with SEHN with US (n = 3), and CT imaging (n = 4).** (A) volume, (B) sphericity, (C) solidity, and (D) SA/V. (E) 3D reconstruction of POL in tissue from US transducer sweep. (F) evaluation of acoustic intensity of POL over 1.5 h. Error bars represent standard deviations of mean value. *p<0.05, and **p<0.01, from t-test statistical test.

Although there was speculation that POL fragment might reach the hepatic veins and lungs (**S6 Fig in S1 Appendix**), no drop in oxygen levels was observed during the injection procedure. Sphericity and solidity showed slight linear correlations with the injected volume of POL from CT imaging (**Fig 7A**), remaining localized over the injection time (**Fig 7B**). The area of the 2D distribution of POL showed linear trends for both US and CT imaging modalities (**Fig 7C**). Major and minor axes of injected POL from both imaging modalities exhibited linear correlations (**Fig 7D and 7E**). Circularity and solidity of injections from CT imaging showed slight linear correlations, while those from US imaging did not (circularity: p = 0.0011, <0.0001, 0.0007, and 0.0045; solidity: (p = 0.0229, <0.0001, 0.0003, and 0.0018). for differences between US and CT imaging at injected volumes of POL 1,2,3, and 4 mL (**Fig 7F** and **7G**). **Fig 7H** depicts the growing 2D distribution of POL over time per mL, indicating the degree of localization throughout the injection. **S5 Table in S1 Appendix**. summarizes 2D and 3D mophometrics per milliliter of POL injected and analyzed using US and CT imaging.

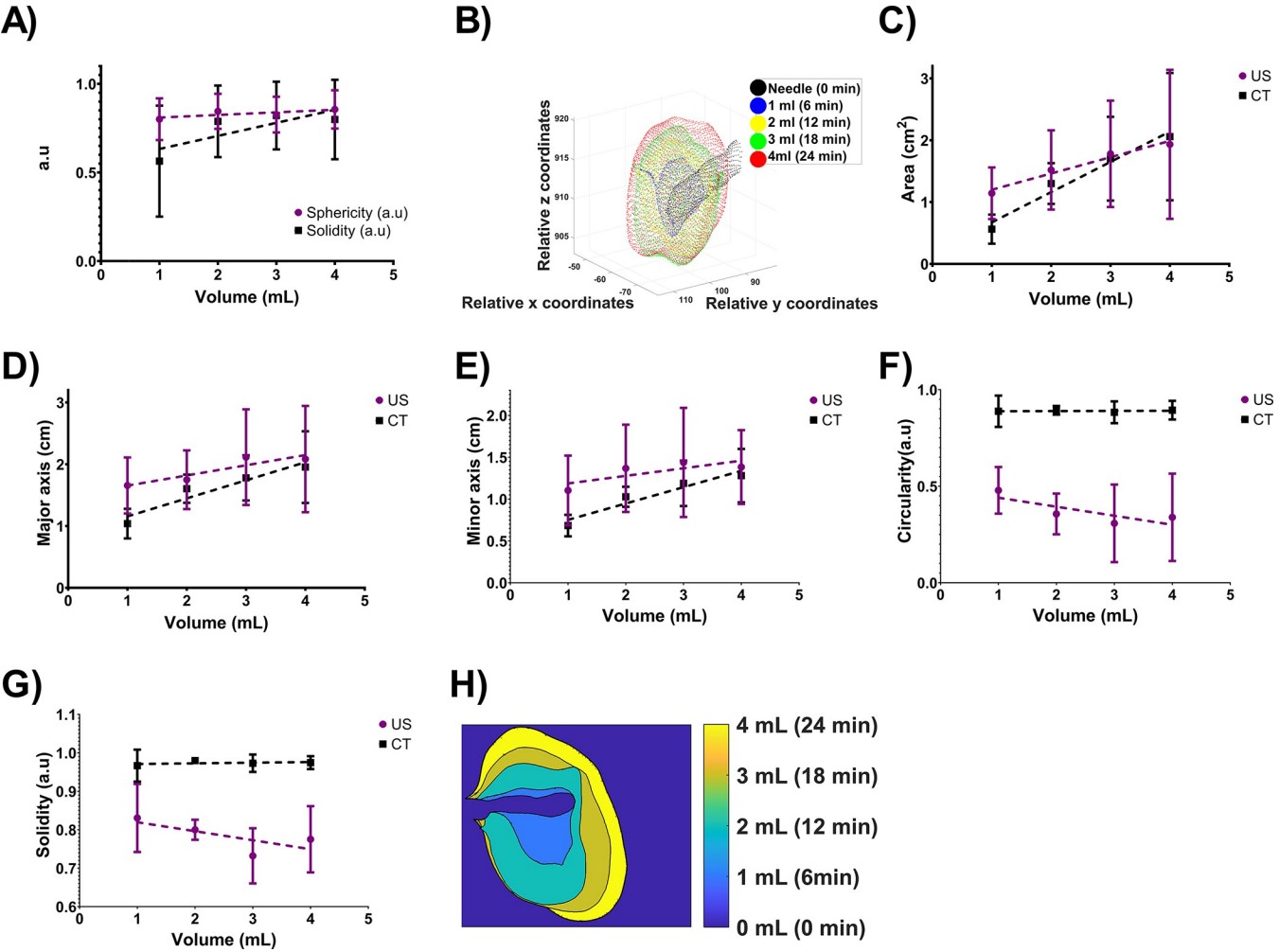

**Fig 7. Morphometrics of POLDOX.** (A) 3D Sphericity and solidity of 4 mL POL injected with SEHN. (B) 3D growing pattern of POL injected at 10 mL/h. (C) Area of POL per mL injected from US and CT imaging. (D) Major axis and minor axis (E) of POL per mL injected from US and CT imaging. (F) Circularity and (G) solidity of POL per mL injected from US and CT imaging. (H) 2D growing pattern of POL per mL injected from CT imaging (coronal plane). Error bars depict standard deviations from mean values (n = 3). Statistical tests were performed with one-way ANOVA.

### Time-course *in vivo* imaging and pharmacokinetic study of poloxamer-based formulation with DOX (POLDOX), and formulation without DOX (DOXSoln)

**CBCT imaging analysis in POLDOX over time.** The 3D morphometrics of the injection pattern of POLDOX was monitored using CBCT imaging (**S7 Fig in S1 Appendix**). The sphericity of POLDOX was not different (p = 0.5368) over the 24 min of injection when compared to all injected volumes (**S7A Fig in S1 Appendix**), and its respective solidity (**S7A Fig in S1 Appendix**) was higher at the first-milliliter injection compared to four-milliliter injection (p = 0.0274). **S7B and S7C Fig in S1 Appendix** depict the 3D distribution of POLDOX over time during injection, with and without leakage, respectively.

Radiopacity on CT imaging of POLDOX was stable over the total evaluated time, 240 min, post-injection (**Fig 8**). There was no difference in Hounsfield units over 240 min, (p = 0.2806) (**Fig 8A**). The volume of POLDOX post-4mL injection at 240 min was different than 0 min (p = 0.0044) and decreased by 26.6% ± 6.4 (**Fig 8B**). The sphericity of POLDOX post-4mL

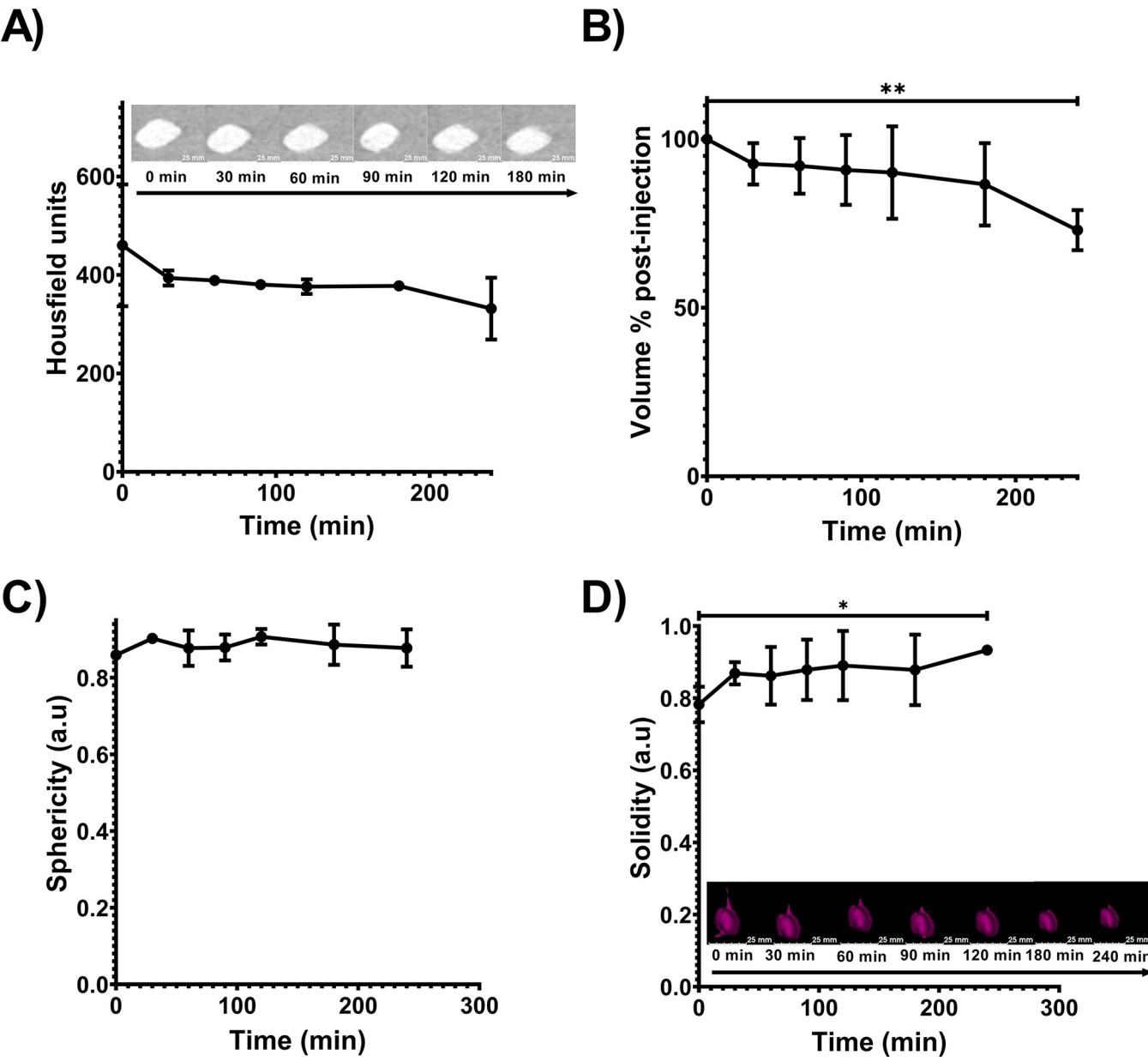

**Fig 8. Characterization of POLDOX with CBCT over time.** (A) Radiopacity of POLDOX post-4mL injection over time. (B) Volume % of POLDOX post-4mL injection over time. (C) Sphericity of POLDOX post-4mL injection over time. (D) Solidity of POLDOX post-4mL injection over time. Error bars depict standard deviations from mean values (n = 3 for 0 min, and n = 2 for 30 min to 240 min). *p<0.05, and **p<0.01, from t-test statistical test when compared values of 0 min and 240 min.

injection from 0 to 240 minutes was no different (p = 0.5368) (**Fig 8C**). Finally, the solidity of POLDOX from 0 min to 240 minutes after 4mL injected increased over time (p = 0.0274) (**Fig 8D**).

Relative iodine concentration over normalized distance were also analyzed over time after POLDOX was fully injected (**S8 Fig in S1 Appendix**). The AUC of relative iodine concentration over normalized distance was not different over time (p>0.9) (**S9 Fig in S1 Appendix**). **S6 Table in S1 Appendix**. summarizes the CBCT imaging analysis in POLDOX over time.

**US imaging analysis of POLDOX over time.** The US acoustic intensity of POLDOX decreased over time (76.8 ±25.9 a.u vs 15.2 ±16.9 a.u, p = 0.0626) for 0 min and 240 min post-

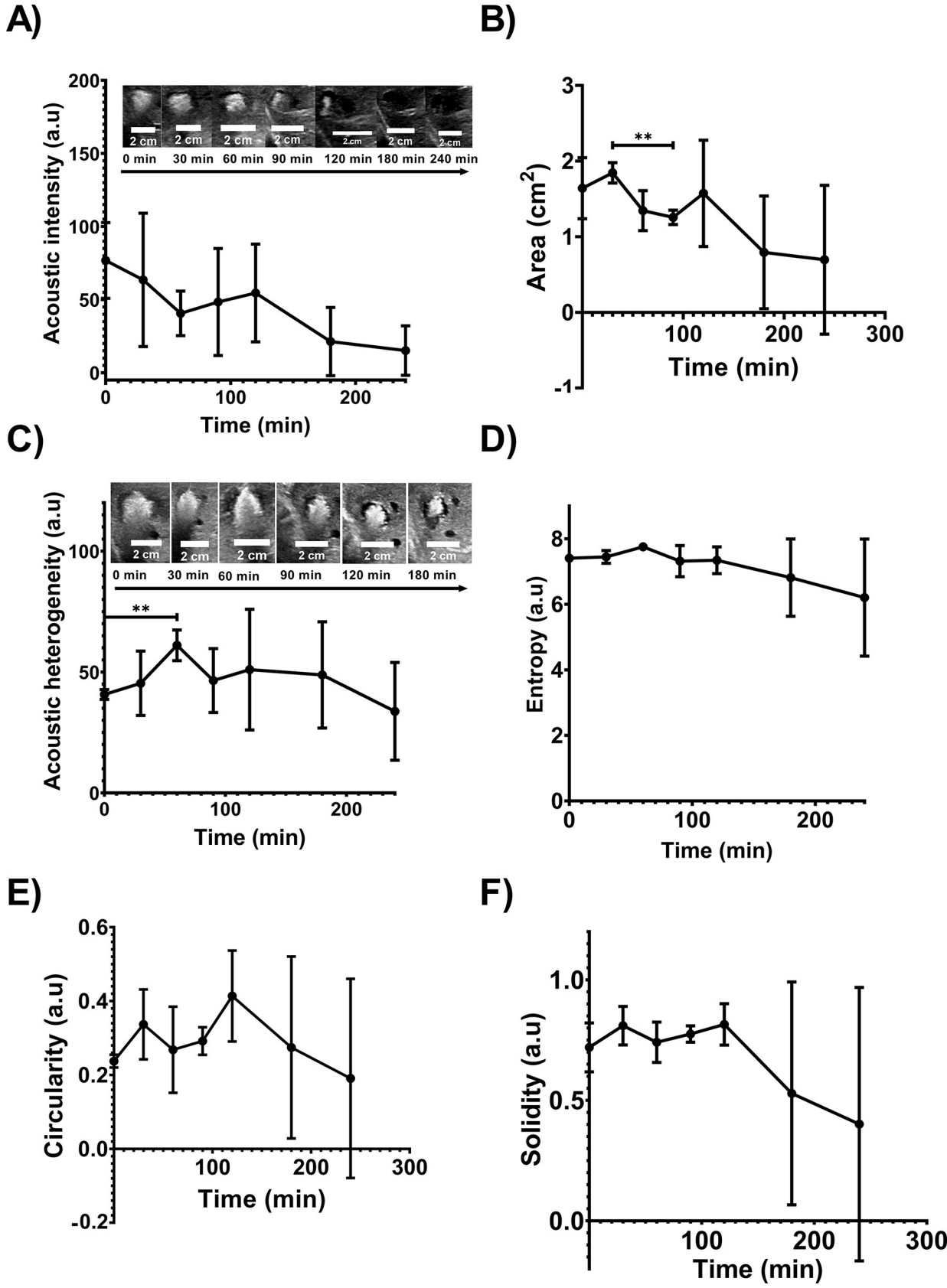

**Fig 9. Characterization of POLDOX with US over time.** (A) Acoustic intensity, (B) area, (C) acoustic heterogeneity, (D) entropy, (E) circularity, and (F) solidity of POLDOX post-4mL-injection over time from US imaging. Error bars depict standard deviations from mean values (n = 3, 0 min to 180 min; n = 2, 240 min). **p<0.01, from t-test statistical study when compared 2 values.

injection, respectively) (**Fig 9A**). The area calculated from US imaging became lower from 1.6 ±0.4cm$^2$ at 0 min to 0.7cm$^2$ ±1.0 at 240 min due to the loss of hyperechoic signal (**Fig 9B**). **S10 Fig in S1 Appendix**, depicts the major and minor axes change over time of POLDOX. The POLDOX deposition over time were hyperechoic post-injection and then hypoechoic regions start to form (**Fig 9A, and 9B**). The acoustic heterogeneity increased over time from 0 min up to 60 min (p = 0.0061) (**Fig 9C**). After 60 min the acoustic heterogeneity tend to decrease (**Fig 9C**). The entropy, defined as a statistical measure of randomness that can be used the texture of an image, was characterized for POLDOX post-4mL injection. The entropy of POLDOX post-4mL injection tended to slightly increase from 0 min to 60 min (p = 0.2925) and then to decrease over time (**Fig 9D**). The circularity of POLDOX from US imaging tended to increase from 0 min up to 120 min and then decrease (**Fig 9E**). Finally, the solidities calculated from US imaging tend to increase from 0 min to 120 min and then decreased (**Fig 9F**). **Table 3** summarizes the US imaging of POLDOX over time.

**CBCT imaging 2D iodine distribution analysis in POLDOX over time.** The ranges of relative iodine concentrations with POLDOX injection were obtained from CBCT imaging (**Fig 10**). The calculated areas of the relative ranges of concentration of iodine were multifaceted (**Fig 10A**). The higher range of iodine concentration, 27 to 40 mg/mL, tended to decrease over time (**Fig 10B**). The area of the middle range of iodine, 13 to 27 mg/mL, concentration tended to increase over time from 0 min to 120 min, and then decreased (**Fig 10B**). Finally, the area of the lower range of iodine concentration, 7 to 13 mg/mL, remained unchanged (**Fig 10B**). The AUC of the areas of relative iodine concentrations over time were calculated and compared over time (**Fig 10C**). The AUC for the highest concentration area over time was bigger (266.8 ±16.5 a.u) compared to the medium (168.6 ±53.2 a.u) (p = 0.1160) and lower (75.6 ±13.7 a.u) (p = 0.0210) areas of concentration ranges of iodine (**Fig 10C**).

The areas of the ranges of iodine concentration were compared per timepoint (**S11 Fig in S1 Appendix**). Overall, at 0 min post-injection, the area of the higher range of iodine concentration remained bigger compared to middle (p = 0.0028) and lower (p = 0.0007) areas of ranges of iodine concentration (**S11A Fig in S1 Appendix**). As time passed, the area of higher range of iodine concentration remained bigger than the other ranges (**S11A-S11C** and **S11E Fig in S1 Appendix**) except for 120 min (**S11D Fig in S1 Appendix**), and 240 min (p>0.4) (**S11F Fig in S1 Appendix**). **Table 4**. Summarizes the areas of relative iodine concentrations in tissue for POLDOX over time.

**Table 3. US imaging analysis and morphometrics of POLDOX.**

| Time post 4mL injection (min) | Acoustic intensity (a.u) | Area (cm$^2$) | Acoustic heterogeneity (a.u) | Entropy (a.u) | Circularity (a.u) | Solidity (a.u) |
|---|---|---|---|---|---|---|
| 0 | 76.8 ±25.9 | 1.6 ±0.4 | 40.7 ±2.0 | 7.4 ±0.1 | 0.2 ±0.0 | 0.7 ±0.1 |
| 30 | 63.5 ±45.6 | 1.8 ±0.1 | 45.3 ±13.3 | 7.4 ±0.2 | 0.4 ±0.1 | 0.8 ±0.1 |
| 60 | 40.6 ±15.2 | 1.3 ±0.3 | 61.1 ±6.3 | 7.7 ±0.1 | 0.4 ±0.1 | 0.7 ±0.1 |
| 90 | 48.4 ±36.6 | 1.2 ±0.1 | 46.5 ±13.2 | 7.3 ±0.5 | 0.3 ±0.0 | 0.8 ±0.0 |
| 120 | 54.5 ±33.4 | 1.6 ±0.7 | 51.0 ±25.0 | 7.3 ±0.4 | 0.4 ±0.1 | 0.8 ±0.1 |
| 180 | 21.3 ±23.3 | 0.8 ±0.7 | 48.8 ±22.0 | 6.8 ±1.2 | 0 ±0.2 | 0.5 ±0.5 |
| 240 | 15.2 ±16.9 | 0.7 ±1.0 | 33.7 ±20.2 | 6.2 ±1.8 | 0 ±0.3 | 0.4 ±0.6 |

n = 3 for 0- min to 180- min values, n = 2 for 240- min values. ± Depicts standard deviations.

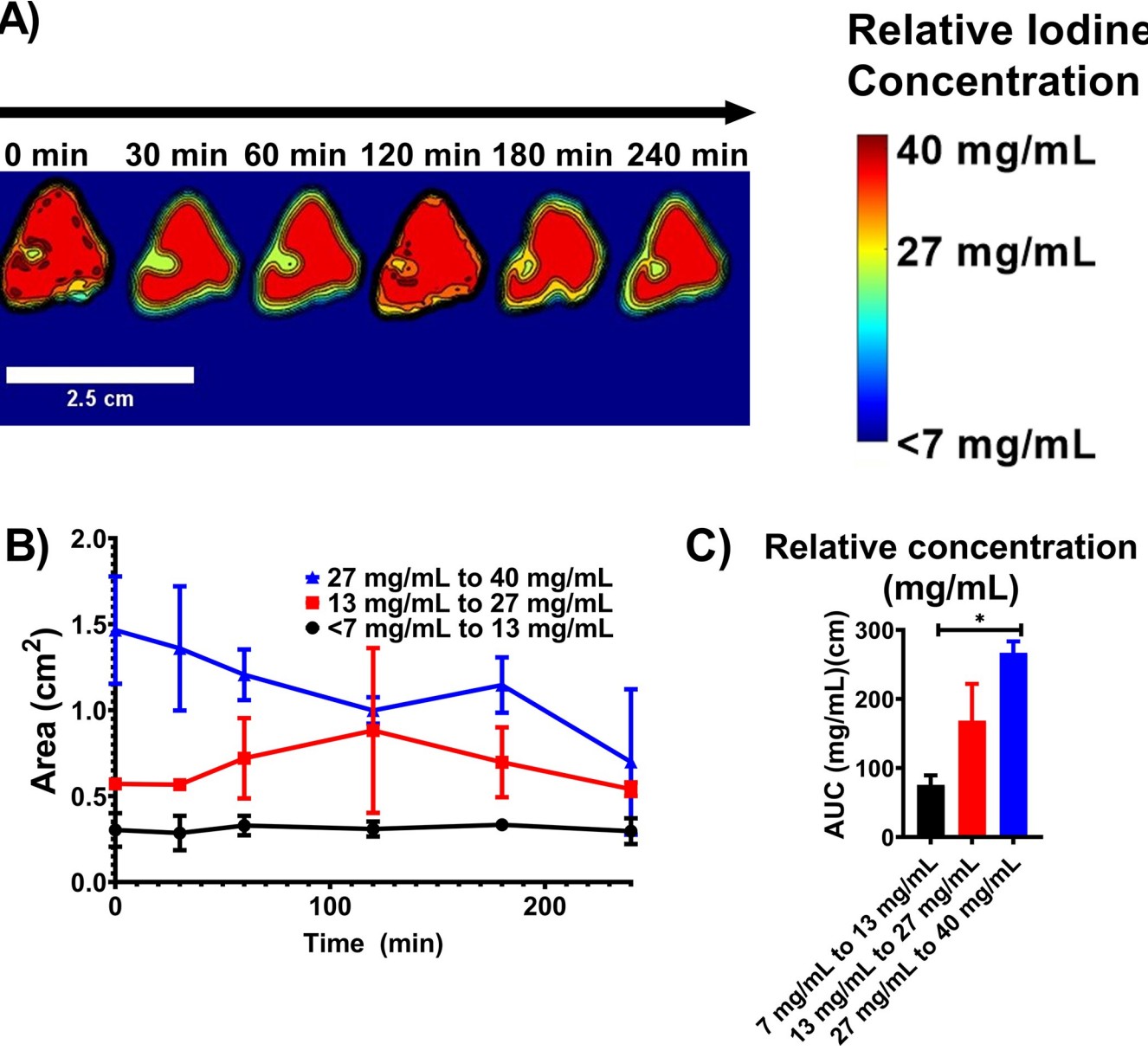

**Fig 10. Iodine distribution over time of POLDOX.** (A) Color-map of the areas of ranges of relative iodine concentration for POLDOX over time. (B) Areas of ranges of relative iodine concentration for POLDOX over time. (C) AUC of areas of ranges of relative iodine concentration for POLDOX from 0 min post-4mL injection to 4h. Error bars depict standard deviations from mean values (n = 3 for 0 min, and n = 2 for 30 min to 240 min). *p<0.05 from one-way ANOVA statistical test.

**3D CBCT and US imaging morphometrical analysis and gross pathology in POLDOX and CBCT morphometrical comparison of POLDOX with DOXSoln.** The CBCT of POLDOX and DOXSoln were also compared (**Fig 11**). POLDOX distributed over a larger contained volume than DOXSoln (**Fig 11A** and**, 11B**) as also shown in fluoroscopy (**Fig 11C**) and gross pathology (**Fig 11D**). POLDOX was able to be easily delineated and was localized compared to DOXSOln (**Fig 11A–11D**) Irregular morphology and high amount of leakage was detected for DOXSoln with CBCT imaging (**Fig 11E**), 3D reconstruction (**Fig 11F**), fluoroscopy (**Fig 11G**), and gross pathology (**Fig 11H**). The sphericity and solidity of POLDOX post-4mL injection were higher (sphericity:0.9 ±0.8 a.u; solidity: 0.8 ±0.5 a.u) than DOXSoln

**Table 4. Areas of relative iodine concentrations in tissue for POLDOX over time.**

| Time post 4mL injection (min) | Relative iodine concentration | | |
|---|---|---|---|
| | 7–13.4 mg/mL | 13.4–27 mg/mL | 27–40 mg/mL |
| 0 | 0.3 ±0.0 | 0.6 ±0.0 | 1.5 ±0.3 |
| 30 | 0.3 ±0.0 | 0.6 ±0.0 | 1.3 ±0.4 |
| 60 | 0.3 ±0.0 | 0.7 ±0.2 | 1.2 ±0.1 |
| 120 | 0.3 ±0.0 | 0.9 ±0.5 | 1.0 ±0.1 |
| 180 | 0.3 ±0.0 | 0.7 ±0.2 | 1.1 ±0.2 |
| 240 | 0.3 ±0.0 | 0.5 ±0.0 | 0.7 ±0.4 |

n = 3 for 0- min values, n = 2 for 30- to 240- min values. ± Depicts standard deviations.

(sphericity:0.8 ±0.0 a.u; solidity: 0.5 ±0.2 a.u), nevertheless, there was not statistical difference (p>0.05) (**Fig 12A**). When dividing the volume of POLDOX and DOXSoln (calculated from CBCT imaging) by 4 mL ($V_{CBCT}$/4 mL), the ratio was higher for POLDOX (0.7 ±0.3) than DOXSoln (0.1 ±0.1) (p = 0.0174) (**Fig 12B**). The sphericity for POLDOX calculated from CBCT and US imaging (CBCT: sphericity = 0.9 ±0.0 a.u; solidity = 0.8 ±0.1 a.u; US: sphericity = 0.8 ±0.0 a.u; solidity = 0.8 ±0.0 a.u) was different (p = 0.0333) (**Fig 12C**). The solidity for POLDOX calculated from CBCT and US imaging was not different (p = 0.3939) (**Fig 12C**).

**Pharmacokinetics of DOX on POLDOX and DOXSoln formulations.** DOX pharmacokinetic (PK) parameters on both formulations were calculated and summarized in **Table 5** and represented on **Fig 13**. On the POLDOX cohort, one animal was excluded from PK data analysis for a final n = 2 because the animal died three hours after treatment. As shown on Fig 13A, DOX was detected in plasma during the administration of both formulations (0 to 24 min), and post-administration over 264 min. However, PK profile showed differences between formulations. $C_{max}$ of DOXSoln was statistically higher compared with POLDOX formulation (5,512 ± 1,658 ng/mL vs 2,084 ± 749 ng/mL, respectively, p = 0.0775), but without significant difference on $T_{max}$. There is a tendency for POLDOX formulation to reach $C_{max}$ later than the DOXSol formulation (25.0 ±1.4 and 21.3 ± 2.3 min for POLDOX and DOXSoln, respectively). DOX exposure to the systemic circulation was also different between formulations. The $AUC_{total}$ of DOXSOln is 2.7-fold higher compared to that of POLDOX (p = 0.0273). A detailed analysis of the AUC showed that it was during the administration time were DOX exposure showed the difference, with $AUC_{administration}$ being 5.3-fold higher for DOXSol compared with POLDOX (p = 0.0194).

The plasma concentration profiles of DOX from both formulations exhibited biexponential decay following completion of the 24 min administration until 264 min (**Fig 13B–13D**). DOX tissue distribution analysis (**Fig 13E**) was performed at the end of the PK study (264 min since administration started). It showed DOX levels still in tested tissues: heart, kidney, spleen, and liver, independently of the formulation administrated but with higher levels on the DOXSol group. Liver was tested on an area far from the site of administration and on the site of administration. Importantly, DOX levels were extremely higher on the site of administration on both formulations compare with the area far from the site of administration. PK data showed that Dox α distribution half-life ($t_{1/2}\alpha$) and β elimination half-life ($t_{1/2}\beta$) [79] are higher in the POLDOX formulation compared with the DOXSol formulation (p = 0.0359, and p = 0.7998 for $t_{1/2}\alpha$ and $t_{1/2}\beta$ respectively), and CL/F rate is higher on POLDOX formulation compared with DOXSol formulation. No DOX was detected in the organs after 4mL administration of POLDOX in one of the swine (animal died); therefore, that animal was excluded from the average DOX quantification in the organs.

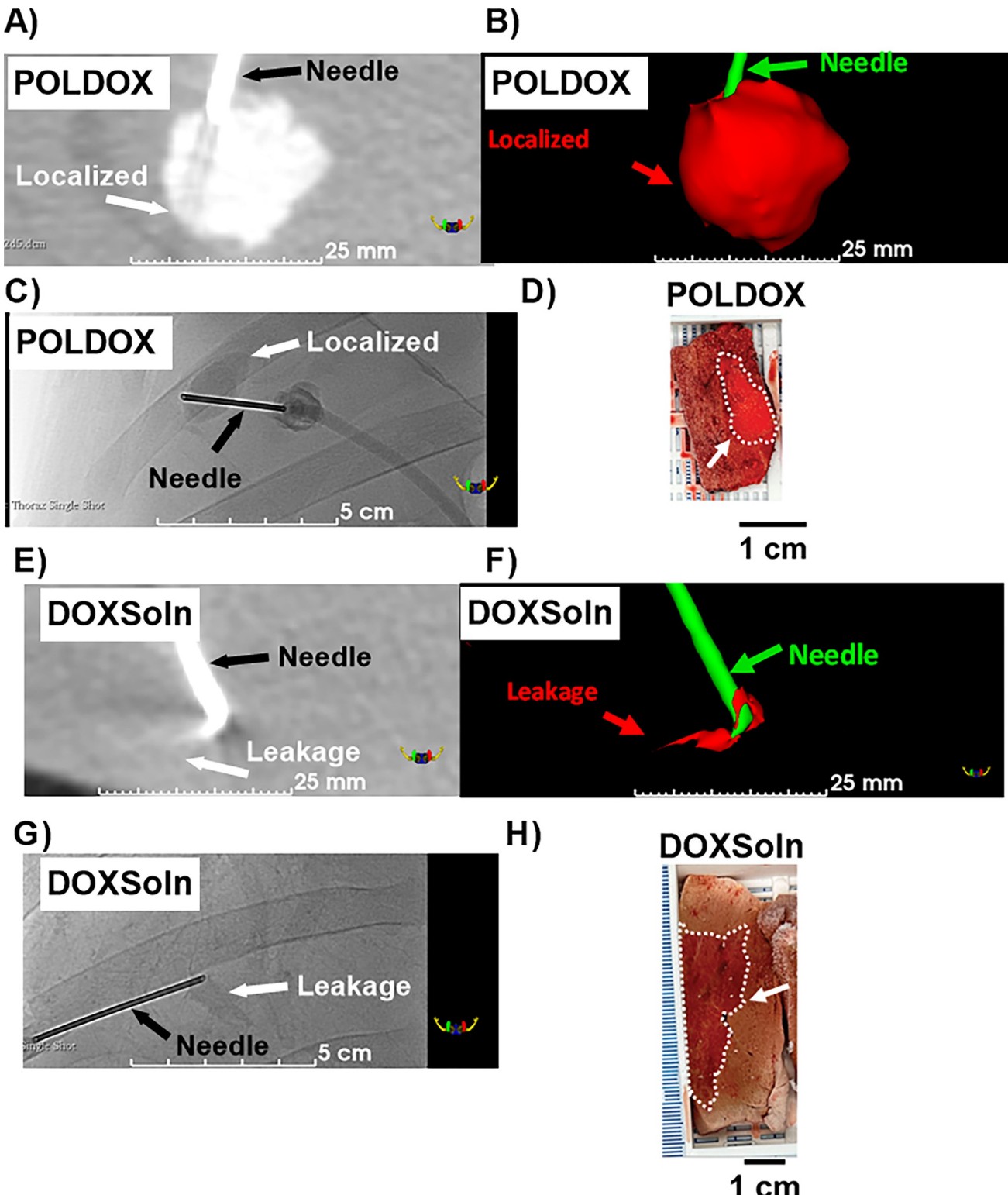

**Fig 11. X-ray imageability of POLDOX and DOXSoln.** (A) CBCT imaging, (B) 3D reconstruction, and (C) fluoroscopy post-administration of POLDOX. (D) Gross pathology of POLDOX after liver explanation. (E) CBCT imaging, (F) 3D reconstruction, and (G) fluoroscopy post-administration of DOXSoln. (H) Gross pathology of DOXSoln after liver explanation.

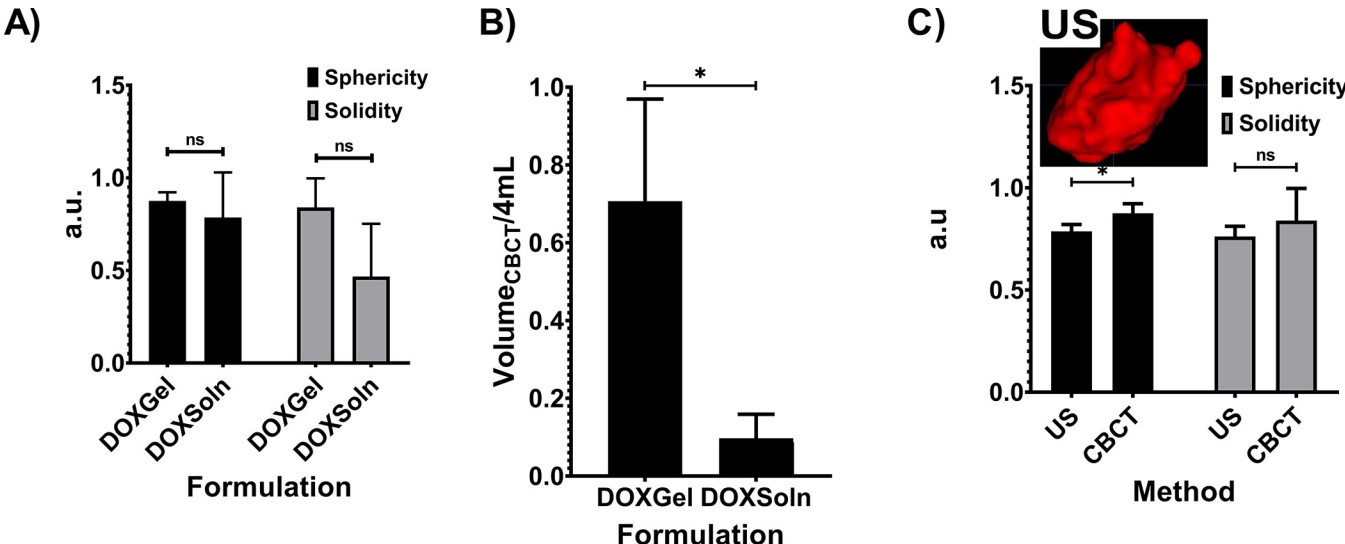

**Fig 12. 3D CBCT and US characterization of POLDOX and DOXSoln.** (A) 3D morphometrical comparison of POLDOX and DOXSoln. (B) Volume $_{CBCT}$/4 mL of POLDOX and DOXSoln. (C) 3D morphometrical comparison of POLDOX from US and CBCT imaging. Error bars depict standard deviations from mean values (n = 3). *p<0.05 from t-test analysis. ns, not significant (p>0.05).

**S7 and S8 Tables in S1 Appendix** summarizes CL/F, DOX levels in plasma for POLDOX, and DOXSoln from the beginning of the administration up to 264 min, and DOX levels in organs and the administration site for POLDOX and DOXSoln at 4h post-4mL POLDOX administration.

## Discussion

This study aimed to assess the feasibility of delivering US and x-ray imageable POL gel formulations using three distinct needle devices in liver, kidney, and muscle tissues in a healthy animal model. It also aimed to delineate the 2D and 3D distribution patterns of POL gels post-delivery via US and x-ray imaging. Finally, pharmacokinetic parameters were computed following percutaneous liver administration of DOX-containing POL gel (POLDOX) in comparison to free DOX administration (DOXSoln).

Insights from these feasibility studies in swine shed light on the deliverability of POL gels via various needle devices, procedural timings, and optimization of MBs concentrations within POL. The visibility on US imaging was contingent upon the concentration of MBs, injection site, and imaging timing.

The 3D morphology of POL, as observed from CT images, exhibited variations based on the needle device used, injection site, and technique. SEHN and MPIN (IM) demonstrated

**Table 5. DOX pharmacokinetic profile from POLDOX (n = 2) and DOXSoln (n = 3) formulations.**

| Formulation | T$_{max}$ (min) | C$_{max}$ (ng/ mL) | AUC$_{administration}$ (ng/ mL·h) | AUC$_{post-administration}$ (ng/ mL·h) | AUC$_{total}$ (ng/ mL·h) | t$_{1/2}$α (min) | t$_{1/2}$β (min) | CL/F ((mg)/(ng/ mL)/h) |
|---|---|---|---|---|---|---|---|---|
| POLDOX | 25.0 ± 1.4 | 2084 ± 749 | 252.4 ± 127.7 | 599.7 ± 281.4 | 852.1 ± 409.1 | 5.4 ± 0.4 | 103.2 ± 24.3 | 0.05 ± 0.02 |
| DOXSoln | 21.3 ± 2.3 | 5512 ± 1658 | 1338.7 ± 304.1 | 944.7 ± 233.5 | 2283 ± 377.2 | 6.8 ± 0.4 | 126.4 ± 111.1 | 0.02 ± 0.00 |
| p values | 0.1456 | 0.0775 | 0.0194 | 0.2285 | 0.0273 | 0.0359 | 0.7998 | 0.0816 |

p values obtained from t-test statistical study. AUC$_{administration}$ and AUC$_{post-administration}$ corresponds to AUC calculated from 0 to 24 min and 24 min to 264 min respectively. AUC$_{total}$ corresponds to AUC calculated from 0 to 264 min.

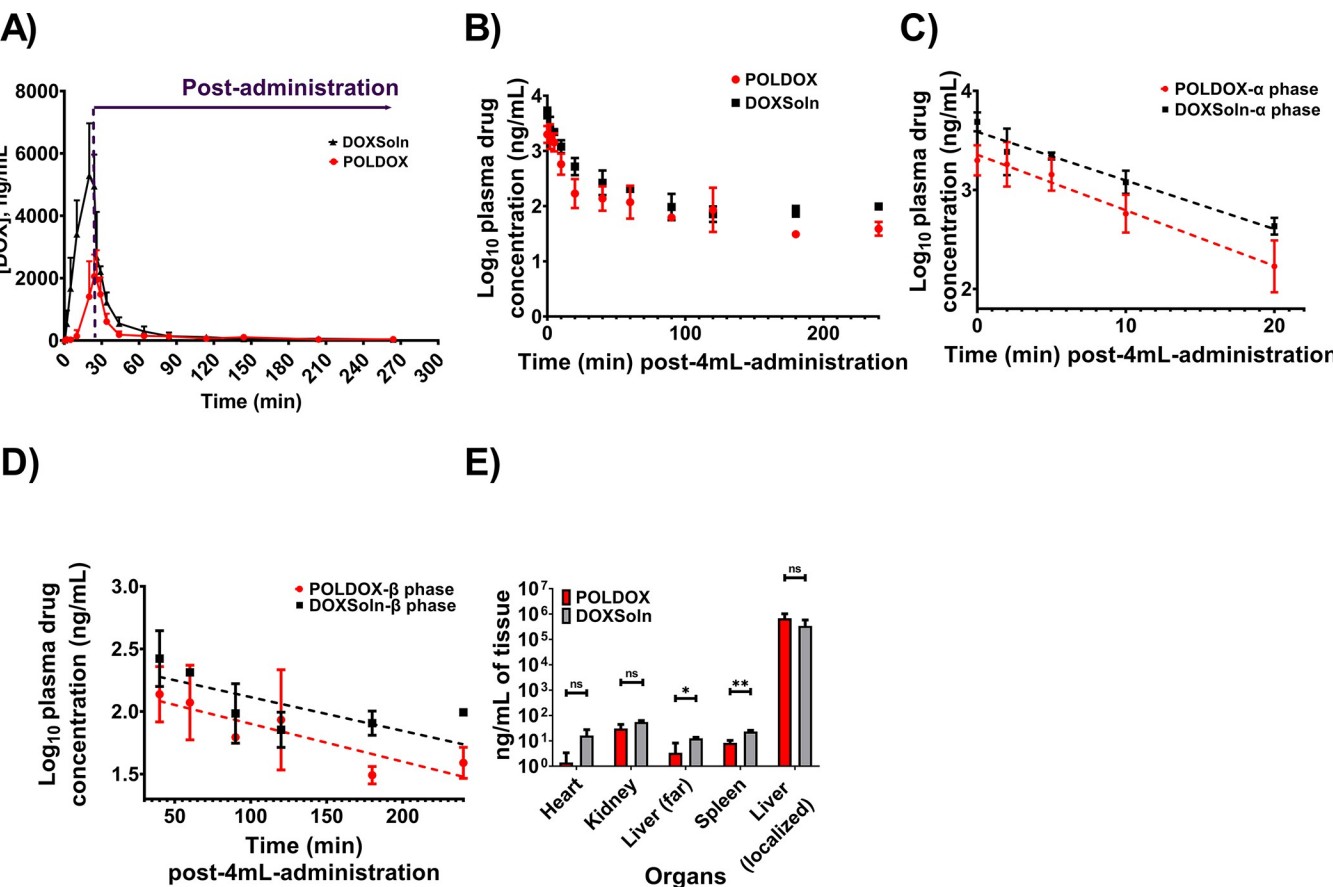

**Fig 13. DOX pharmacokinetic profile from POLDOX and DOXSoln formulations.** (A) DOX plasma in plasma concentration over time during and after 4mL formulation administration. Each dot represent mean ± SD, n = 2 for POLDOX cohort and n = 3 for DOXSol cohort. (B) Log$_{10}$ of DOX concentration over time post-4mL-administration of POLDOX and DOXSoln. (C) α distribution and (D) β elimination phases of Log$_{10}$ of DOX concentration over time post-4mL-administration of POLDOX and DOXSoln. (E) Biodistribution of DOX at the administration site and organs for POLDOX and DOXSoln. Error bars depict standard deviations from mean values for POLDOX (n = 3), and for DOXSoln (n = 3) for (A). Error bars depict standard deviations from mean values for POLDOX (n = 2), and for DOXSoln (n = 3) for (B, C, and D). For the case of biodistribution, error bars depict standard deviations from mean values (n = 2, in organs; n = 2 for administration site in the liver) for POLDOX, and for DOXSoln (n = 3). Not significant, ns, p>0.05, *p<0.05, and **p<0.01, from t-test statistical test when compared POLDOX and DOXSoln. n, number of animals. For plotting Fig 13B, the time notation for 4mL- post-administration is considered from the end of the administration (0 min) to 240 min.

consistent local delivery of POL injected into the liver and in muscles, respectively, compared to MSHN and MPIN devices used in the liver. SEHN's reliability attributed to less chaotic fluid advection [80] during gel transport, while MPIN (IM) benefited from injection in low-vascularity paraspinous muscle regions. Conversely, multiple holes in MSHN and multiple prongs in MPIN resulted in decreased predictability although injection site is a factor on injection localization. These findings suggest that SEHN had the highest reproducibility in terms of gel distribution within the injection site, while MSHN exhibited less reproducible gel distribution in healthy tissue. MPIN showed more reproducibility than MSHN, with concentrations corresponding to needle tip locations. Injection in the kidney were also reproducible and localized in 2 out of three attempts, nevertheless, morphologies were not as spherical due to the anatomical physical barriers of the kidney. For example, the kidney medulla region, served as a mold filled with POL which impacted its morphology. Overall, understanding these distribution patterns and tissue characteristics can aid in optimizing needle device selection for precise delivery of substances like iodine during medical procedures.

The most homogeneous and most concentrated relative iodine concentration was observed in SEHN injections which may be due to the formation of tissue cavities filled with POL compared to the other needle devices [81]. The assessment of AUC from relative mg/mL of iodine over normalized distance demonstrated consistent results with SEHN and MPIN (IM).

3D morphological parameters depended on the needle device and injection technique. SA/V can be intentionally modulated, *in vivo*. SEHN provided spherical shapes which will result in more morphological predictability, nevertheless, the SA/V control will be limited compared to MPIN. MPIN provided the opportunity to modulate SA/V from the selection of different needle retractions and rotations which may impact pharmacokinetic parameters; changes in surface area will affect biodegradability of the polymer and therefore impact rate of drug release [82]. This observation is aligned with a previous study performed on *ex vivo* bovine liver where SA/V could be modulated depending on needle devices and injection techniques [9]. These findings highlight the differences in injection reproducibility, distribution patterns, and morphometric characteristics among different needle devices, which can inform decision-making regarding their use in POL injections for various medical applications.

The visibility of POL gel under US imaging may be influenced by injection pressures, with SEHN demonstrating clearer circular shapes compared to MSHN and MPIN. Increase in injection pressure as a function of needle gauge was previously observed [30] and may be a reason of MBs destruction in MSHN and MPIN injections. The comparison of 2D morphometrics from CT and US were overall comparable demonstrating that both imaging modalities may be used to assess treatment zones. Lower circularity values in SEHN measurements may be explained due to the transducer pushing on the subject's skin, which may disturb original gel morphology compared to CT imaging. From this observation, physicians will need to take into consideration that interrogating treatment zones with US may display disturbed surrounding tissue which may also disturb gel morphology.

3D structures obtained from US imaging using electromagnetic tracking systems offered the possibility to evaluate treatment volumes without the use of ionizing radiation. The volumes in US tended to be larger than sizes measured in CT. In general, CT imaging provided more consistent results as evidenced with lower standard deviation values and covariances of morphometrical measurements compared to US imaging. This indicates that CT imaging may be more precise and robust for a more accurate prediction of treatment volumes of POL compared to US. However, some of the disadvantages of CT imaging are the cost and the use of ionizing radiation during clinical procedures. However, the combination of US and x-ray imaging modalities may reduce procedural times as well as the exposure of ionizing energy compared to CT or CBCT alone.

The evaluation of the 3D distribution of POL that leaked to adjacent vessels needs further investigation. This study showed that POL leakage trajectory could be determined by CT imaging. For a particular example of a leakage event in this study, a POL trajectory is speculatively directed towards the hepatic vein. Therefore, adverse effects of these organs need to be further explored. US imaging evaluation of leaked material will not be as readily visible compared to CT imaging in the abdomen due to the interference of thoracic cage bones on the US beam. The serial imaging of POL injection per milliliter over time was useful to determine potential drug-gel leakage during the injection procedure. However, serial imaging in the clinical setting may not be feasible as it is important to consider potential radiation exposure to patients.

Lower solidity values of POLDOX 3D distribution during injection may be an indicative of future systemic exposure of DOX while high levels of sphericity and solidity suggested localized POLDOX distribution and therefore minimal systemic exposure.

The 3D analysis of POLDOX morphometrics post-injection provided information on drug transport over the interrogated time with CBCT imaging. Reduction of volume over time is

indicative of potential drug diffusion in the tissue and POLDOX interface [83, 84]. The 3D solidity over time tended to increase which may be an indication of the transport of POLDOX filling vessels at the surface of the injection site. At initial stages the solidity is lower due to the effect of leaking material filling vessels, product of gel intravasation (wire-like shapes), as visualized in the surface of the gel deposition, as time goes by these wire-like materials of gel filling vessels disappear potentially due to the convective transport from diffusion (concentration driven) and blood bulk velocity transport (advection) [83] resulting in smooth surfaces of POLDOX.

Interestingly the US imageability post-injection of POLDOX had a more complex set of characteristics. The acoustic intensity and area of deposited material tended to be hyperechoic and then converted to hypoechoic due potentially to MBs diffusion. This may explain prominent standard deviations in morphometric parameters, towards the end to the study. As an empirical observation, POLDOX formulations tended to maintain imageability for longer periods of time compared to POL without DOX. It was observed that POL without DOX over time produced hypoechoic cores surrounded by hyperechoic halos, by contrast, POLDOX over time provided hypoechoic halos encapsulating hyperechoic regions. This may be explained by the addition of hydrophobicity from DOX into POL formulation which impedes amphiphilic micelles being eroded by water. The acoustic heterogeneity and entropy measurements of POLDOX demonstrated the complexity of MBs diffusion. At the initial stage of US imageability post-injection, the acoustic heterogeneity and entropy tended to be lower compared to the 60 min timepoint, this may be due to homogenous acoustic signal distributed within the injection site at initial stages (bright POL depots). As time goes by, MBs diffused creating regions with hypo and hyperechoic acoustic characteristics increasing acoustic heterogeneity and entropy until finally these two parameters dropped due to major regions of hypoechoic homogeneity (black POL depots).

Relative iodine concentrations were also assessed in POLDOX with a clear tendency of an area reduction associated with the highest range of relative concentration, 27 mg/mL to 40 mg/mL, due potentially to diffusion transport. The increase of the middle relative iodine concentration range associated area (13 mg/mL to 27 mg/mL) at the 120 min timepoint is likely due to the diffusion effects of molecule transport from higher concentrations to lower concentration gradients [83]. The lower range of iodine concentration area remained constant suggesting that drug transport will be mainly though diffusion transport from the localized POL depot towards tissue (assuming no POL located in vessels). Finally, the AUC of the areas of the relative concentration of iodine over time demonstrated that the POLDOX imageability remained stable and therefore visible over the tested timepoints.

Fluoroscopic imaging and CBCT of POLDOX compared to DOXSoln demonstrated a high level of localization due to the incorporation of POL into DOX formulation. Notably, most of the volume for the POLDOX formulation was contained within the tissue 3D space compared to DOXSoln. This may signify a more effective delivery of DOX formulation and potentially fewer side effects. The comparison 3D morphometrics of POLDOX from US and CBCT imaging were similar, therefore it is possible to predict 3D treatment zones with US imaging reducing ionizing radiation.

During the injection of POLDOX and DOXSoln, it was detected DOX amounts which may be attributed to rapid intravasation of drug compared to POLDOX as observed in US and CBCT imaging. The comparison of solidities using CBCT and US imaging from intravasated POLDOX during serial growth analyses may be used also to suggest DOX presence during injection.

Pharmacokinetic analysis provided insight into DOX disposition following percutaneous intrahepatic administration of POLDOX and DOXSoln formulations. In one of the swine

treated with POLDOX, DOX levels were not detected in either the plasma or the examined organs post-4mL administration. This absence was attributed to the localization of POLDOX at the injected site without any leakage, as confirmed by CBCT imaging. Unfortunately, this animal also died after 3 hours of treatment. Post-mortem examination suggested a hypertrophic cardiomyopathy, a known occurrence in swine which is associated with sudden death, and likely unrelated to DOX toxicity. As a result, for the POLDOX cohort plasma pharmacokinetic assessment, only data from the two animals were used to calculate $T_{max}$, $C_{max}$, $t_{1/2}$s, and CL/F. The DOX $T_{max}$ from DOXSoln formulation was expected to occur earlier than for the POLDOX formulation due to the rapid intravasation of the low-viscosity material (DOXSoln). In addition, the PK analysis showed a tendency for the $T_{max}$ from POLDOX to be longer (p = 0.1456) perhaps because a significant portion of DOX is encapsulated within the hydrogel of the POLDOX formulation, whereas free DOX in DOXSoln tends to bind to the tissue where it resides during the experimental timeframe. CT images of POLDOX and DOXSoln (**Fig 11A and 11E**) and gross pathology (**Fig 11G and 11H**) confirmed substantial DOX localization at the POLDOX injection site and retention in the vicinity of DOXSoln injection. This may mitigate systemic exposure of DOX levels post-injection, likely attributed to drug retention within the gel.

The lower $C_{max}$ and $AUC_{tot}$ of POLDOX compared to DOXSol were consistent with reduced systemic exposure that may be attributed to drug retention within the gel. (**Table 5**, **and S7 Table in S1 Appendix**). $AUC_{tot}$ is reported here as the sum of $AUC_{administration}$, and $AUC_{post-administration}$. A lower AUC for POLDOX (**Fig 13A and Table 5 and S7 Table in S1 Appendix**) indicates that less DOX is available in the bloodstream over the course of the study compared to DOXSoln. This observation is supported by data showing higher concentrations of POLDOX at the injection site compared to DOXSoln (**Fig 11A, 11E, 11D and 11H**). These findings suggest that POLDOX remains more localized at the injection site and is less distributed throughout the body. In simpler terms, the lower $AUC_{tot}$ for POLDOX occurs because the drug does not spread widely through the bloodstream. Instead, it stays concentrated at the injection site. This localization might explain why the drug appears to clear from the bloodstream faster—not because it is removed from the body more quickly, but because less of it is present in the bloodstream due to its high concentration at the injection site.

Retention in the liver is further supported by the finding that one swine had no detectable DOX in the plasma post-4mL POLDOX administration, although this animal had pre-existing health issues as found post-mortem. Further studies over a prolonged duration are necessary to fully characterize the pharmacokinetic profile of this formulation. DOX PK profile from both formulations showed to undergo enterohepatic circulation showed by the bi-exponential plasma profile [55, 85, 86]. As the DOX is highly localized at the administration site, $t_{1/2}\alpha$ and $t_{1/2}\beta$ values for POLDOX were longer compared to DOXSoln (p = 0.0359, and p = 0.7998) which appears as DOX being present at a shorter time after POLDOX administration. Nevertheless, as explained for the CL/F values, POLDOX formulation is cleared from plasma faster as its $AUC_{total}$ is lower, meaning there was less amount of DOX circulating into the system compared to DOXSoln formulation. The fact that POLDOX formulation allowed lower DOX systemic exposure than DOXSoln formulation (**Fig 13**) suggests POLDOX as an efficient platform for local drug delivery.

The amount of DOX in the heart after administration with POLDOX compared to DOXSoln was 8-fold lower which may translate to a reduction of DOX-related cardiotoxicity risk or enabling a larger local dose for the same level of cardiac exposure. This assumption is based on the premise that much of the cardiotoxicity is from the cumulative DOX dose [87], as with congestive heart-failure [88], and arrhythmogenic effects [89].

POLDOX exhibited a substantial reduction in drug accumulation in the kidney, spleen, and distant portions of the liver by 2-, 3-, and 4-fold respectively compared to DOXSoln. This emphasizes the efficacy of the gel in delivering anti-cancer agents more effectively to the site of injection. Additionally, gross pathology findings indicated greater localization of POLDOX near the injection site, supporting the reduction in systemic exposure to DOX compared to DOXSoln.

This study has several important limitations that should be considered. Firstly, the pharmacokinetic analysis was limited to four hours following completion of drug injections and relied on a small animal number. Increasing the animal number is essential to ensure the reliability and applicability of the study's findings to broader clinical contexts. Secondly, the use of healthy animal models and healthy tissue may not accurately reflect how drugs distribute in a tumor environment. Therefore, future research should incorporate animal models with tumors to demonstrate therapeutic efficacy and explore potential combination therapies more accurately. This approach will provide insights that are more relevant to clinical scenarios involving cancer treatment. Thirdly, gel injections displace tissue by physically pushing aside or displacing surrounding tissue. The differential exit, or differential interstitial pressures or fascial planes can result in uneven distribution of gel within tissues or tumors, which poses major morphometric and drug distribution challenges for consistent drug delivery. To thoroughly evaluate the gel's suitability for clinical use: i) long-term survival studies are essential to comprehensively assess the gel's performance and biodegradability over extended periods, ii) detailed histopathological analysis is crucial to understand how the gel interacts with tissue, providing insights into its effects and potential improvements for future investigations. Lastly, further efficacy and toxicological investigations should encompass a diverse range of drugs, such as small molecules, proteins, or monoclonal antibodies used in chemotherapy or immunotherapy. These studies need to evaluate potential issues like drug leakage into adjacent vessels.

Addressing these limitations is crucial to enhance the understanding and broaden the applicability of gel-based drug delivery systems in clinical settings. This comprehensive approach will provide valuable insights into the safety and efficacy of these systems across various therapeutic contexts, thereby advancing their potential for clinical use.

## Conclusions

This investigation explores the utilization of POL-based hydrogels as a carrier for drug delivery, incorporating US and CT or CBCT contrast agents into the formulations. The study demonstrated the feasibility of delivering POL using three needle devices under image guidance for intramuscular, renal, and hepatic delivery. Tunable POL distribution was achieved by intentionally varying injection techniques intentionally. The inclusion of multimodal imaging into POL formulations allowed for navigation systems coupled with electromagnetic tracking, facilitating the prediction of treatment zones in 3D from US imaging and minimizing exposure to ionizing radiation. This approach may optimize clinical workflows by reducing intervention procedure times. Morphometrics obtained from imaging modalities in the context of pharmacokinetic behavior were used to characterize the behavior of the gel compared to free drug in solution. This information assessed comparable exposures and potential toxicities across vital organs based on that distribution. Overall, injections of POL containing DOX (POLDOX) resulted in lower systemic drug exposure compared to injections of free DOX (DOXSoln). Moreover, a higher amount of DOX was found in organs following DOXSoln administration compared to POLDOX. This favorable profile may elucidate the rationale behind local drug delivery benefits over systemic exposures.

This research supports the use of hydrogels as potential drug delivery vectors for image-guided interstitial injections, enhancing drug delivery precision with more favorable therapeutic and reduced systemic exposure compared to traditional intravenous administration of anti-cancer agents. The addition of dual imageability alongside of an enhanced local to systemic profile supports further work and translation of such drug plus device paradigms under image-guided injections across diseases like cancer.

## Supporting information

**S1 Appendix. Materials and methods, image processing code, and supplementary figures.**
(DOCX)

**S2 Appendix. Raw data used to calculate pharmacokinetic parameters and DOX biodistribution.**
(XLSX)

## Acknowledgments

J.F.D. is currently a participant of the Graduate Partnership Program through the University of Maryland Fischell Department of Bioengineering in the A. James Clark School of Engineering. The authors are in debt to Dr. Huang Chiao Huang, Dr. Jenna Mueller, Dr. Helim Espinoza Aranda from University of Maryland, College Park for their useful discussion. The authors are also grateful to Dr. Cecile Dufour from Philips in Paris for assistance in the reconstruction of 3D US sweeps. The authors are also grateful to Lindsey Hazen, Scientific Program Coordinator for the Center for Interventional Oncology, for helping to coordinate this project. We are in debt to with Dr. Veronica Ramirez Alcantara from the University of South Alabama for her insightful feedback pertaining to pharmacokinetics.

## Author Contributions

**Conceptualization:** Jose F. Delgado, Nicole A. Varble, Andrew S. Mikhail, Joshua W. Owen, William F. Pritchard, Bradford J. Wood.

**Data curation:** Jose F. Delgado, Ayele H. Negussie, Nicole A. Varble, Ivane Bakhutashvili, William F. Pritchard.

**Formal analysis:** Jose F. Delgado, Ayele H. Negussie, Bradford J. Wood.

**Funding acquisition:** Bradford J. Wood.

**Investigation:** Jose F. Delgado, Ayele H. Negussie, Antonio Arrichiello, Tabea Borde, Laetitia Saccenti, Ivane Bakhutashvili, Robert Morhard, William F. Pritchard.

**Methodology:** Jose F. Delgado, Ayele H. Negussie, Nicole A. Varble, Andrew S. Mikhail, Antonio Arrichiello, Joshua W. Owen, John W. Karanian, William F. Pritchard, Bradford J. Wood.

**Project administration:** John W. Karanian, William F. Pritchard, Bradford J. Wood.

**Resources:** Bradford J. Wood.

**Software:** Nicole A. Varble.

**Supervision:** Ayele H. Negussie, William F. Pritchard, Bradford J. Wood.

**Validation:** Jose F. Delgado, Ayele H. Negussie, William F. Pritchard.

**Visualization:** Jose F. Delgado.

**Writing – original draft:** Jose F. Delgado.

**Writing – review & editing:** Jose F. Delgado, Ayele H. Negussie, Nicole A. Varble, Andrew S. Mikhail, Antonio Arrichiello, Tabea Borde, Laetitia Saccenti, Ivane Bakhutashvili, Joshua W. Owen, Robert Morhard, John W. Karanian, William F. Pritchard, Bradford J. Wood.

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
