## [Decision Letter · Decision Letter 0]

1 Oct 2024

PONE-D-24-37693In vivo Imaging and Pharmacokinetics of Percutaneously Injected Ultrasound and X-ray Imageable Thermosensitive Hydrogel loaded with Doxorubicin versus Free Drug in SwinePLOS ONE

Dear Dr. Delgado,

Thank you for submitting your manuscript to PLOS ONE. After careful consideration, we feel that it has merit but does not fully meet PLOS ONE’s publication criteria as it currently stands. Therefore, we invite you to submit a revised version of the manuscript that addresses the points raised during the review process.

We look forward to receiving your revised manuscript.

Kind regards,

Isha Mutreja

Academic Editor

PLOS ONE

Journal Requirements: When submitting your revision, we need you to address these additional requirements. 1. Please ensure that your manuscript meets PLOS ONE's style requirements, including those for file naming. The PLOS ONE style templates can be found at https://journals.plos.org/plosone/s/file?id=wjVg/PLOSOne_formatting_sample_main_body.pdf and https://journals.plos.org/plosone/s/file?id=ba62/PLOSOne_formatting_sample_title_authors_affiliations.pdf 2. Please note that PLOS ONE has spec6ific guidelines on code sharing for submissions in which author-generated code underpins the findings in the manuscript. In these cases, all author-generated code must be made available without restrictions upon publication of the work. Please review our guidelines at https://journals.plos.org/plosone/s/materials-and-software-sharing#loc-sharing-code and ensure that your code is shared in a way that follows best practice and facilitates reproducibility and reuse. 3. We note that the grant information you provided in the ‘Funding Information’ and ‘Financial Disclosure’ sections do not match.  When you resubmit, please ensure that you provide the correct grant numbers for the awards you received for your study in the ‘Funding Information’ section. 4. Thank you for stating the following financial disclosure: "“This work was supported by the NIH Center for Interventional Oncology and the Intramural Research Program of the National Institutes of Health via intramural NIH Grants Z1A CL040015, 1ZIDBC011242." Please state what role the funders took in the study.  If the funders had no role, please state: ""The funders had no role in study design, data collection and analysis, decision to publish, or preparation of the manuscript."" If this statement is not correct you must amend it as needed. Please include this amended Role of Funder statement in your cover letter; we will change the online submission form on your behalf. 5. Thank you for stating the following in the Acknowledgments Section of your manuscript: "This work was supported by the Center for Interventional Oncology in the Intramural Research Program of the National Institutes of Health (NIH) by intramural NIH Grants NIH Z01 1ZID BC011242 and CL040015 and by the National Cancer Institute grant 1ZIABC011313-09. J.F.D. was supported by the Jayne Koskinas  Ted Giovanis Foundation for Health and Policy through the NIH Graduate Partnership Program from the  Office of Intramural Training and Education (GPP-OITE). J.F.D. is currently a participant of the Graduate Partnership Program through the University of Maryland Fischell Department of Bioengineering in the A. James Clark School of Engineering." We note that you have provided funding information that is not currently declared in your Funding Statement. However, funding information should not appear in the Acknowledgments section or other areas of your manuscript. We will only publish funding information present in the Funding Statement section of the online submission form. Please remove any funding-related text from the manuscript and let us know how you would like to update your Funding Statement. Currently, your Funding Statement reads as follows: "“This work was supported by the NIH Center for Interventional Oncology and the Intramural Research Program of the National Institutes of Health via intramural NIH Grants Z1A CL040015, 1ZIDBC011242." Please include your amended statements within your cover letter; we will change the online submission form on your behalf. 6. Thank you for stating the following in the Competing Interests section: "BW is Principal Investigator on the following CRADA’s = Cooperative Research & Development Agreements, between NIH and industry: Philips (CRADA), Philips Research (CRADA), Celsion Corp (CRADA), BTG Biocompatibles / Boston Scientific) (CRADA), Siemens (CRADA), NVIDIA (CRADA), XAct Robotics (CRADA). ProMaxo (CRADA), Negotiating CRADA with Tempus, Galvanize, Theromics, Imactis, Canon Medical, Varian, MediView. NV is a staff Scientist from Philips Healthcare." Please confirm that this does not alter your adherence to all PLOS ONE policies on sharing data and materials, by including the following statement: ""This does not alter our adherence to  PLOS ONE policies on sharing data and materials.” (as detailed online in our guide for authors http://journals.plos.org/plosone/s/competing-interests).  If there are restrictions on sharing of data and/or materials, please state these. Please note that we cannot proceed with consideration of your article until this information has been declared.  Please include your updated Competing Interests statement in your cover letter; we will change the online submission form on your behalf. 7. PLOS requires an ORCID iD for the corresponding author in Editorial Manager on papers submitted after December 6th, 2016. Please ensure that you have an ORCID iD and that it is validated in Editorial Manager. To do this, go to ‘Update my Information’ (in the upper left-hand corner of the main menu), and click on the Fetch/Validate link next to the ORCID field. This will take you to the ORCID site and allow you to create a new iD or authenticate a pre-existing iD in Editorial Manager. 8. We note that you have included the phrase “data not shown” in your manuscript. Unfortunately, this does not meet our data sharing requirements. PLOS does not permit references to inaccessible data. We require that authors provide all relevant data within the paper, Supporting Information files, or in an acceptable, public repository. Please add a citation to support this phrase or upload the data that corresponds with these findings to a stable repository (such as Figshare or Dryad) and provide and URLs, DOIs, or accession numbers that may be used to access these data. Or, if the data are not a core part of the research being presented in your study, we ask that you remove the phrase that refers to these data. 9. Please include captions for your Supporting Information files at the end of your manuscript, and update any in-text citations to match accordingly. Please see our Supporting Information guidelines for more information: http://journals.plos.org/plosone/s/supporting-information. 10. Please review your reference list to ensure that it is complete and correct. If you have cited papers that have been retracted, please include the rationale for doing so in the manuscript text, or remove these references and replace them with relevant current references. Any changes to the reference list should be mentioned in the rebuttal letter that accompanies your revised manuscript. If you need to cite a retracted article, indicate the article’s retracted status in the References list and also include a citation and full reference for the retraction notice.

Reviewers' comments:

Reviewer's Responses to Questions

**Comments to the Author**

1. Is the manuscript technically sound, and do the data support the conclusions?

Reviewer #1: Yes

2. Has the statistical analysis been performed appropriately and rigorously? 

Reviewer #1: Yes

3. Have the authors made all data underlying the findings in their manuscript fully available?

Reviewer #1: No

4. Is the manuscript presented in an intelligible fashion and written in standard English?

Reviewer #1: Yes

5. Review Comments to the Author

Reviewer #1: The manuscript titled “In vivo Imaging and Pharmacokinetics of Percutaneously Injected Ultrasound and X-ray Imageable Thermosensitive Hydrogel loaded with Doxorubicin versus Free Drug in Swine” is well written and highlights the potential application of Intratumoral injection of injectable hydrogel assisted by ultrasound and X-ray CT imaging for localized delivery to address the adverse side effects as a result of systemic overdose. The work is interesting but prior to acceptance, I would suggest adding information in the Materials and Methods section for reproducing the results/study in relation to lipid microbubbles and image analysis using MATLAB and ImageJ.

Materials and Methods:

1. Page 5: Please include a complete list of chemicals and reagents used during the study that should include the product codes and company names.

2. Page 6: line 147-148, what is the final volume of aqueous phase?

3. Page 6: line 154, Volume of POL 22% used?

4. Page 6: line 156, volume of saline and volume of DOX 10mg/ml used?

5. Page 6: line 158-160, volume of each component?

6. Page 10, please provide program script for MATLAB and conditions used for Fiji for image analysis in supplemental information.

6. PLOS authors have the option to publish the peer review history of their article (what does this mean?). If published, this will include your full peer review and any attached files.

Reviewer #1: No

---

## [Author Response · Author response to Decision Letter 0]

20 Oct 2024

Dear Dr. Isha Mutreja,

Academic Editor

PLOS ONE

We greatly appreciate your time in considering our manuscript with Submission ID PONE-D-24-37693 titled “In vivo Imaging and Pharmacokinetics of Percutaneously Injected Ultrasound and X-ray Imageable Thermosensitive Hydrogel loaded with Doxorubicin versus Free Drug in Swine”. Your guidance and the reviewer’s suggestions were highly appreciated to improve this manuscript. 

Please find below point by point (in blue color) the responses for each point you kindly provided.

Thank you very much for considering our manuscript.

Sincerely,

Bradford J. Wood, MD.

Center for Interventional Oncology

Clinical Center

National Institutes of Health

Jose F. Delgado, PhD. Cand.

Center for Interventional Oncology

Clinical Center

National Institutes of Health

Fischell Department of Bioengineering

University of Maryland, College Park

Answer: The manuscript was adapted to the suggested PLOSOne format.

2. Please note that PLOS ONE has spec6ific guidelines on code sharing for submissions in which author-generated code underpins the findings in the manuscript. In these cases, all author-generated code must be made available without restrictions upon publication of the work. Please review our guidelines at https://journals.plos.org/plosone/s/materials-and-software-sharing#loc-sharing-code and ensure that your code is shared in a way that follows best practice and facilitates reproducibility and reuse.

Answer: Thank you for bringing this point, the codes were included in S1 Apprendix.

 Answer: We incorporated the grant numbers in the Financial disclosure section of the manuscript.

"“This work was supported by the NIH Center for Interventional Oncology and the Intramural Research Program of the National Institutes of Health via intramural NIH Grants Z1A CL040015, 1ZIDBC011242."

 Answer: The role of funders was added as follows in the funding disclosure section: "The funders had no role in study design, data collection and analysis, decision to publish, or preparation of the manuscript."

"This work was supported by the Center for Interventional Oncology in the Intramural Research Program of the National Institutes of Health (NIH) by intramural NIH Grants NIH Z01 1ZID BC011242 and CL040015 and by the National Cancer Institute grant 1ZIABC011313-09. J.F.D. was supported by the Jayne Koskinas Ted Giovanis Foundation for Health and Policy through the NIH Graduate Partnership Program from the Office of Intramural Training and Education (GPP-OITE). J.F.D. is currently a participant of the Graduate Partnership Program through the University of Maryland Fischell Department of Bioengineering in the A. James Clark School of Engineering."

"“This work was supported by the NIH Center for Interventional Oncology and the Intramural Research Program of the National Institutes of Health via intramural NIH Grants Z1A CL040015, 1ZIDBC011242."

Answer: We removed the funding information from acknowledgements and add it to financial disclosure section.

6. Thank you for stating the following in the Competing Interests section: 

"BW is Principal Investigator on the following CRADA’s = Cooperative Research & Development Agreements, between NIH and industry: Philips (CRADA), Philips Research (CRADA), Celsion Corp (CRADA), BTG Biocompatibles / Boston Scientific) (CRADA), Siemens (CRADA), NVIDIA (CRADA), XAct Robotics (CRADA). ProMaxo (CRADA), Negotiating CRADA with Tempus, Galvanize, Theromics, Imactis, Canon Medical, Varian, MediView. NV is a staff Scientist from Philips Healthcare."

Answer: “We added the following information to the conflicts of interest section: "This does not alter our adherence to PLOS ONE policies on sharing data and materials.”

We also removed: Uro 1 (CRADA), and Mediview (CRADA). Negotiating CRADA with Tempus, Galvanize, Theromics, Imactis, Canon Medical, Varian, MediView.

7. PLOS requires an ORCID iD for the corresponding author in Editorial Manager on papers submitted after December 6th, 2016. Please ensure that you have an ORCID iD and that it is validated in Editorial Manager. To do this, go to ‘Update my Information’ (in the upper left-hand corner of the main menu), and click on the Fetch/Validate link next to the ORCID field. This will take you to the ORCID site and allow you to create a new iD or authenticate a pre-existing iD in Editorial Manager.

Answer: Thank you, the ORCID iD for the corresponding author was updated.

8. We note that you have included the phrase “data not shown” in your manuscript. Unfortunately, this does not meet our data sharing requirements. PLOS does not permit references to inaccessible data. We require that authors provide all relevant data within the paper, Supporting Information files, or in an acceptable, public repository. Please add a citation to support this phrase or upload the data that corresponds with these findings to a stable repository (such as Figshare or Dryad) and provide and URLs, DOIs, or accession numbers that may be used to access these data. Or, if the data are not a core part of the research being presented in your study, we ask that you remove the phrase that refers to these data.

Answer: Thank you for bringing this important note, the following information was modified in the manuscript:

Line 514. “No drop in oxygen levels was observed during the injection procedure.”

 Answer: The captions were added for supporting information files.

 Answer: The reference list was reviewed and they are correct.

Dear Reviewer,

We thank you and the reviewers for an in-depth review of our manuscript (Manuscript Submission ID PONE-D-24-37693) and now submit a revised version addressing the reviewers’ comments in this letter and manuscript. We sincerely thank the reviewers and Editor for their thoughtful and constructive comments that resulted in an improved manuscript by correcting specific weaknesses. The manuscript has now been modified according to the reviewers’ recommendations. The specific modifications are listed below in a point-by-point manner.

We appreciate the reviewers’ kind comments which make the team feel good about the work. Thoughtful words included: “the manuscript is well written and highlights the potential application of Intratumoral injection……..” and “The work is interesting……”.

We thank you for your time and consideration of this revised manuscript.

Sincerely,

Bradford J. Wood, MD.

Center for Interventional Oncology

Clinical Center

National Institutes of Health

Jose F. Delgado, PhD. Cand.

Center for Interventional Oncology

Clinical Center

National Institutes of Health

Fischell Department of Bioengineering

University of Maryland, College Park

Reviewer #1: The manuscript titled “In vivo Imaging and Pharmacokinetics of Percutaneously Injected Ultrasound and X-ray Imageable Thermosensitive Hydrogel loaded with Doxorubicin versus Free Drug in Swine” is well written and highlights the potential application of Intratumoral injection of injectable hydrogel assisted by ultrasound and X-ray CT imaging for localized delivery to address the adverse side effects as a result of systemic overdose. The work is interesting but prior to acceptance, I would suggest adding information in the Materials and Methods section for reproducing the results/study in relation to lipid microbubbles and image analysis using MATLAB and ImageJ.

Materials and Methods:

1. Page 5: Please include a complete list of chemicals and reagents used during the study that should include the product codes and company names.

Thank you for your suggestion. A complete list of chemicals and reagents used during this study were incorporated in S1 Appendix, page 1.

We incorporated the following statement in the manuscript. 

Page 5, Lines 134 to 135.

“The complete list of chemicals, and reagents used during this study are listed below and included in supplemental information as S1 Appendix, page 1.”S1 Appendix, page 1“.

Materials and methods

Chemicals.

1,2-distearoyl-sn-glycero-3-phosphocholine (18:0 DSPC) (850365C-1g; Avanti lipids, Alabaster, Alabama, USA) and 0.4mL of 10mg/mL of Polyethylene glycol (PEG) 40 stearate (P3440-250G), both in chloroform (372978-1L) (Sigma Aldrich, Inc; Saint Louis, MO, USA). Normal saline (114-055-101, Quality Biological, Gaithersburg, Maryland, USA), glycerol (G7757-500ml), and propylene glycol (81380-1L) (Sigma Aldrich). Perfluorobutane gas (001836-100G) was obtained from Matrix Scientific, Columbia, SC, USA. Poloxamer 407, purified non-ionic (16758-250G) was obtained from Sigma Aldrich. Iodixanol (1123771), Visipaque 320 mg/mL, was obtained from GE Healthcare, Waukesha, WI, USA. Doxorubicin, Hydrochloride Salt, >99% (D-4000) was obtained from LC Laboratories, Woburn, Massachusetts, USA. Intramuscular ketamine (NC-0256) (25 mg/Kg), and midazolam (NC-0534) (0.5 mg/Kg) were obtained from Next Gen, Weatherford, Texas. Glycopyrrolate (0.01 mg/Kg) (NDC 0517-4620-25, American Regent, INC, Shirley, New York, USA) and anesthetized with propofol (1 mg/Kg intravenous) (A18293, Adooq Bioscience LLC, Irvine, California, USA) and maintained under general anesthesia with isoflurane (1-5%, Isoflo, Abbott Animal Health, North Chicago, IL). Beuthanasia-D (pentobarbital sodium 390 mg/mL and phenytoin sodium 50 mg/mL) (NDC 0061-0473-05) was obtained from Merck, New Jersey, USA. 2-mercaptoethanol was obtained from Sigma Aldrich. 0.1% trifluoroacetic acid (AC325330100) was obtained from Thermo Fisher Scientific, Waltham, Massachusetts), and acetonitrile (100665) was obtained from Sigma Aldrich. Daunorubicin (1164700-200MG) and Doxorubicin (1225703-50 MG) for calibration standards were obtained from Sigma Aldrich. Potassium phosphate monobasic (KH₂PO₄) and Zinc Sulfate monohydrate (ZnSO4.H2O) were obtained also from Sigma Aldrich.

2. Page 6: line 147-148, what is the final volume of aqueous phase?

We appreciate the feedback; the manuscript has been updated as follows:

Page 6, Lines 147 to 148.

“The phospholipid film was solubilized in a 2 mL solution of normal saline, glycerol, and polypropylene glycol in an 8:1:1 ratio and heated at 79°C for 1 hour.

3. Page 6: line 154, Volume of POL 22% used?

The volume updated as follows,

Page 6, Lines 154 to 157.

“POL gel was prepared as previously reported (38). A 22% (w/v) formulation of POL (Sigma Aldrich) containing 40mg/mL of iodine from iodixanol (Visipaque 320 mg/mL, GE Healthcare, Waukesha, WI, USA) was solubilized in normal saline (110 g of POL, 62.5 mL of iodixanol, and 327.5 mL of normal saline) at 4°C under vigorous stirring for 24h”.

4. Page 6: line 156, volume of saline and volume of DOX 10mg/ml used?

updated as follows

Page 6, Lines 157 to 159.

Similarly, POL containing iodixanol and 10mg/mL of DOX (LC Laboratories, Woburn, MA, USA) was prepared under the same conditions (27.5 mg of POL, 15.6 mL of iodixanol, 1.25 g of DOX, and 109.4 mL of normal saline) and stirred vigorously for 120h until complete dissolution was achieved.

5. Page 6: line 158-160, volume of each component?

updated as follows

Page 6, Lines 162 to 163.

“One microliter of MBs was added to 10 mL of POL solution to achieve a concentration of 0.01% MBs (v/v), while ten microliters were added to achieve a concentration of 0.1% MBs (v/v), with or without DOX. “

6. Page 10, please provide program script for MATLAB and conditions used for Fiji for image analysis in supplemental information.

Thank you for the suggestion. We included the following MATLAB codes and ImageJ analysis in the supplemental information. Additionally, we added a statement in the manuscript to help readers locate this information.

Page 11, Lines 277 to 278.

MATLAB codes, and Image J methodology cab be found at supplemental information (S1 Appendix).

S1 Appendix, page 1 to 3.

“

Matlab codes:

Calculation of area, circularity, and solidity

close all

clear all

I=imread('Slide4.jpg');

I=rgb2gray(I);

imshow(I)

th = graythresh(I);

IcropEqTh = im2bw(I,th);

for i = 1

 Icrop = imcrop(IcropEqTh) ;

 imshow(Icrop)

 stats(:,i) = regionprops(Icrop,'Area','Perimeter','Solidity','MajorAxisLength','MinorAxisLength','Orientation','Circularity');

end

Calculation of entropy

% Load the image (Replace 'your_image.jpg' with your actual image file)

img = imread('Slide23.jpg');

% Convert to grayscale if the image is RGB

if size(img, 3) == 3

 img_gray = rgb2gray(img);

else

 img_gray = img;

end

% Display the image and select ROI manually using a polygon

figure;

imshow(img_gray);

title('Draw a polygon around the ROI and double click when done');

roi = drawpolygon('Label', 'Select ROI', 'Color', 'r');

% Wait for the ROI to be finalized

wait(roi);

% Create a binary mask for the ROI

mask = createMask(roi);

% Extract the ROI using the mask

selectedROI = img_gray;

selectedROI(~mask) = 0; % Set pixels outside the ROI to 

---

## [Decision Letter · Decision Letter 1]

12 Nov 2024

In vivo Imaging and Pharmacokinetics of Percutaneously Injected Ultrasound and X-ray Imageable Thermosensitive Hydrogel loaded with Doxorubicin versus Free Drug in Swine

PONE-D-24-37693R1

Dear Dr. Delgado,

We’re pleased to inform you that your manuscript has been judged scientifically suitable for publication and will be formally accepted for publication once it meets all outstanding technical requirements.

Kind regards,

Isha Mutreja

Academic Editor

PLOS ONE

Additional Editor Comments (optional):

Reviewers' comments:

Reviewer's Responses to Questions

**Comments to the Author**

1. If the authors have adequately addressed your comments raised in a previous round of review and you feel that this manuscript is now acceptable for publication, you may indicate that here to bypass the “Comments to the Author” section, enter your conflict of interest statement in the “Confidential to Editor” section, and submit your "Accept" recommendation.

Reviewer #1: (No Response)

2. Is the manuscript technically sound, and do the data support the conclusions?

Reviewer #1: (No Response)

3. Has the statistical analysis been performed appropriately and rigorously? 

Reviewer #1: (No Response)

4. Have the authors made all data underlying the findings in their manuscript fully available?

Reviewer #1: (No Response)

5. Is the manuscript presented in an intelligible fashion and written in standard English?

Reviewer #1: (No Response)

6. Review Comments to the Author

Reviewer #1: (No Response)

7. PLOS authors have the option to publish the peer review history of their article (what does this mean?). If published, this will include your full peer review and any attached files.

Reviewer #1: No

---

## [Editor Report · Acceptance letter]

27 Nov 2024

PONE-D-24-37693R1 

PLOS ONE

Dear Dr. Delgado, 

I'm pleased to inform you that your manuscript has been deemed suitable for publication in PLOS ONE. Congratulations! Your manuscript is now being handed over to our production team.

Kind regards, 

on behalf of

Dr. Isha Mutreja 

Academic Editor

PLOS ONE